# Sex-specific behavioral feedback modulates sensorimotor processing and drives flexible social behavior

Sarath Ravindran Nair[1], Adrián Palacios-Muñoz [1,2,7], Sage Martineau[2,3], Malak Nasr[2,4,5] & Jan Clemens [1,2,6] ✉

How the brain enables individuals to adapt behavior to their partner is key to understanding social exchange. For example, courtship behavior involves sensorimotor processing of signals that can result in behavioral dialog between partners, such as stereotyped movements and singing. The courtship behavior of *Drosophila melanogaster* males with their partners, which are usually female but can also be male, involves singing. To investigate how behavioral feedback and sensorimotor processing contribute to flexible social interactions, we compared the courtship behavior and singing of male *D. melanogaster* towards males and females. Quantitative analysis of their interactions revealed that while underlying courtship and song rules are unaffected by the sex of the partner, the behavioral dynamics and song sequences differ by partner sex. This divergence stems from sex-specific behavioral feedback: females decelerate to song, while males orient towards the singer. Moreover, optogenetic manipulations reveal that the partners' responses are driven by sex-specific neural circuits that link song detection with arousal and social decisions. Our findings demonstrate that flexible social behaviors can arise from fixed sensorimotor rules through a context-dependent selection facilitated by the partner's behavioral feedback. More broadly, our results reveal compositionality as a key mechanism for achieving behavioral flexibility during complex social interactions such as courtship.

The brain enables flexible behavior by dynamically transforming sensory cues into motor outputs through sensorimotor rules[1–5]. These rules are not necessarily fixed[6–10], but flexibly selected and modulated based on the organism's internal state and the external sensory context[11–16]. In social settings, each individual's behavior generates sensory cues for others, creating a closed loop of reciprocal influence. To navigate dynamic social environments, the brain must interpret and flexibly respond to social signals in real time. This flexibility can be achieved by applying different rules for different partners—based on innate preferences[17–20], learning[21–23], or prior experience[24–26]—and by modulating behavior in response to varying social feedback when applying each rule[13,27]. Understanding the mechanisms underlying flexible rule selection and application is key to unraveling how the brain controls flexible behavior.

Courtship interactions are an example of a behavior shaped by behavioral feedback and sensorimotor processing. Although typically directed at members of the opposite sex, same-sex courtship is also widespread in the animal kingdom (see ref. 28 for a comprehensive

[1]ENI-G, a Joint Initiative of the University Medical Center Göttingen and the Max Planck Institute for Multidisciplinary Sciences, Göttingen, Germany. [2]IMPRS Neuroscience, Göttingen, Germany. [3]Institute of Computer Science, University of Göttingen, Göttingen, Germany. [4]CERVO Brain Research Centre, Québec City, QC, Canada. [5]Faculty of Medicine, Université Laval, Québec City, QC, Canada. [6]Department of Neuroscience, Faculty VI, University of Oldenburg, Oldenburg, Germany. [7]Present address: Charité Universitätsmedizin Berlin, Berlin, Germany. ✉e-mail: jan.clemens@uol.de

review). Homosexual courtship is often assumed to be identical to heterosexual courtship behavior, but the extent to which individuals employ the same rules for homosexual versus heterosexual courtship is unclear. Importantly, members of the same sex often respond differently to being courted than members of the opposite sex[29–31], yet whether this differential feedback shapes the behavior of the courter remains poorly understood. Thus, comparing courtship behaviors directed at different sexes offers a compelling entry point for understanding how the brain produces flexible behavior through the interplay of behavioral rules and feedback[1,5,32–34].

In *Drosophila melanogaster*, chemical cues indicate the sex and species of an individual[35–37]. Males sample the chemical profile of other flies and typically initiate courtship toward females of their own species[36,38,39]. The male chases the female and produces a courtship song by extending and vibrating one wing. This courtship song contains two main modes[40]: pulse song, consisting of regular trains of two types of short pulses[41], and sine song, a sustained oscillation of the wing. The sensorimotor rules that transform feedback from females to song patterning in the courter determine the timing, duration, and composition of the song[4,6,13,41,42]. This involves males switching between a set of three rules, *close*, *chasing*, and *whatever*, depending on the male's internal state and interaction context[13]. Males use the *whatever* rule when not interested in an interaction, the *chasing* rule when the female is fast and distant to produce mainly pulse song, and the *close* rule when she is slow and nearby, producing mainly sine song.

Male-male interactions in *Drosophila* in the presence of a resource —food or a female—are often aggressive and involve head butting, boxing, fencing, wing threats, lunging, and chasing[43–45]. During aggressive interactions, males produce agonistic song, which is more irregular than courtship song and generated by bilateral wing flicking[43–47]. However, males also frequently court other males in the wild[48]. In laboratory settings, courtship-like behavior between males can be induced via genetic mutations but can also occur spontaneously. In groups, song playback leads to courtship-like chaining behavior[25,49–51], during which males chase each other's tails and extend one wing. However, whether the unilateral wing extensions during male-male interactions produce song, and if so, whether male-directed song comprises the same modes and patterns as female-directed song, has not been systematically examined.

By comparing male- and female-directed courtship and modes of singing, we elucidate how social feedback and internal sensorimotor rules interact to generate flexible social behavior. We find that male- and female-directed courtship unfolds with different dynamics. This results in differences in song patterns, which arise because of the partners' sex-specific behavior: Female partners typically stop during courtship, while male partners often turn back to face the courting male. This turning behavior in males leads to different sensory experiences from singing to females, which changes when specific rules are used and thus singing behavior. Moreover, we identify a putative neural circuit that links song perception with different behavioral responses in males. Taken together, our results demonstrate how flexible behavior emerges from compositionality, such that fixed sensorimotor rules are used according to social feedback. Our study thus proposes a mechanism through which flexibility can be achieved even in innate behaviors such as courtship.

## Results

### Male- and female-directed courtship-like interactions exhibit different dynamics

To compare male- and female-directed singing, we tracked the interactions of male-male and male-female pairs[52] and recorded acoustic signals with an array of 16 microphones[4,53,54]. To promote male-male interactions, we controlled the males' prior social experience by housing them in groups and perfumed the chamber with male and female flies prior to the experiments (see Methods). In our assay,

group-housed males interacted intensely with both males and females (females 86 ± 12% vs. males 73 ± 15%, Fig. 1A). Female-directed interactions had lower latency and were more frequent, indicating that the males still discriminated between sexes (Fig. 1A). Using uni- and bilateral wing extension as proxies of courtship and aggression, we found that even the male-male interactions were courtship-like: Males primarily displayed unilateral wing extensions, with bilateral extensions being nearly absent in male-female pairs and rare in male-male pairs (Fig. 1B). The audio recordings confirmed our interpretations of the wing extensions: Males produced mainly courtship song when interacting with either sex, while agonistic song was absent during male-female interactions and rare during male-male interactions (Fig. 1B).

However, although male- and female-directed interactions were both courtship-like, they unfolded with different dynamics. During male-female interactions, the male spent 84 ± 12% of the time behind the female, facing her tail. By contrast, male-male pairs frequently (42 ± 31%) engaged in head interactions, with both males facing each other (Figs. 1C, D, S1). These head interactions were not specific to the NM91 wild-type strain we used in the assay: OregonR wild-type males also engaged in frequent head-directed interactions with male but not female partners (males: 14 ± 10% vs females: 7 ± 5%, Fig. S1). Although head interactions between males typically indicate aggression[45,55], males produced primarily courtship song during the head interactions (Fig. S3A), suggesting that these encounters were courtship-like rather than aggressive.

To more comprehensively compare the dynamics of male- and female-directed interactions, we generated social maps using UMAP, which embeds the egocentric and relational kinematics of both flies into two dimensions (Figs. 1E, F, S2)[56–59]. In these maps, similar interaction patterns are positioned close together, whereas dissimilar patterns are separated. High-density regions correspond to stereotypical interaction patterns (social modes), while low-density regions indicate transitions between modes. Because UMAP is a nonlinear embedding method, distances and the shapes of high-density regions are not directly interpretable. The embedding space was therefore smoothed using kernel density estimation and segmented into social modes using watershed-based spatial segmentation. The social maps identified eight behaviorally relevant social modes (Figs. 1F, G, S2, Supplementary Videos 1–8) and revealed both shared and sex-specific interaction patterns with male and female partners (Fig. 1F, H): Males spent similar time chasing male or female partners ("behind chasing"). However, during male-female interactions, males were more often sitting "behind close" or in front of the female ("front idle")[60]. By contrast, males were more often oriented towards each other and close ("front close"). Males engaged in head interactions with both males and females, and the head interactions differed not only in frequency (Fig. 1H) but also in their dynamics (Fig. 1I). During head interactions with a female, males were more distant and either idle or circled around a stationary female. During male-directed head interactions, males were closer and moved more slowly (Fig. 1I). Therefore, the dynamics of male- and female-directed courtship-like interactions are different, thus creating distinct sensory experiences for a courting male.

### Song patterns depend on the partner's sex during head interactions

Given that courtship song is shaped by the sensory experiences of the male[4,6,13,41,42], we investigated whether male- and female-directed song patterns have distinct properties, particularly during head interactions. Song is patterned on two timescales: On a short timescale (<50 ms), the frequency content of the sine song and pulses as well as the interval between pulses are produced by largely hard-wired circuits in the ventral nerve cord[61]. Accordingly, male- and female-directed songs were nearly identical on the short timescale, with similar carrier frequencies of pulse and sine, pulse durations, intervals, and waveform shapes (Figs. 2A, B, S3).

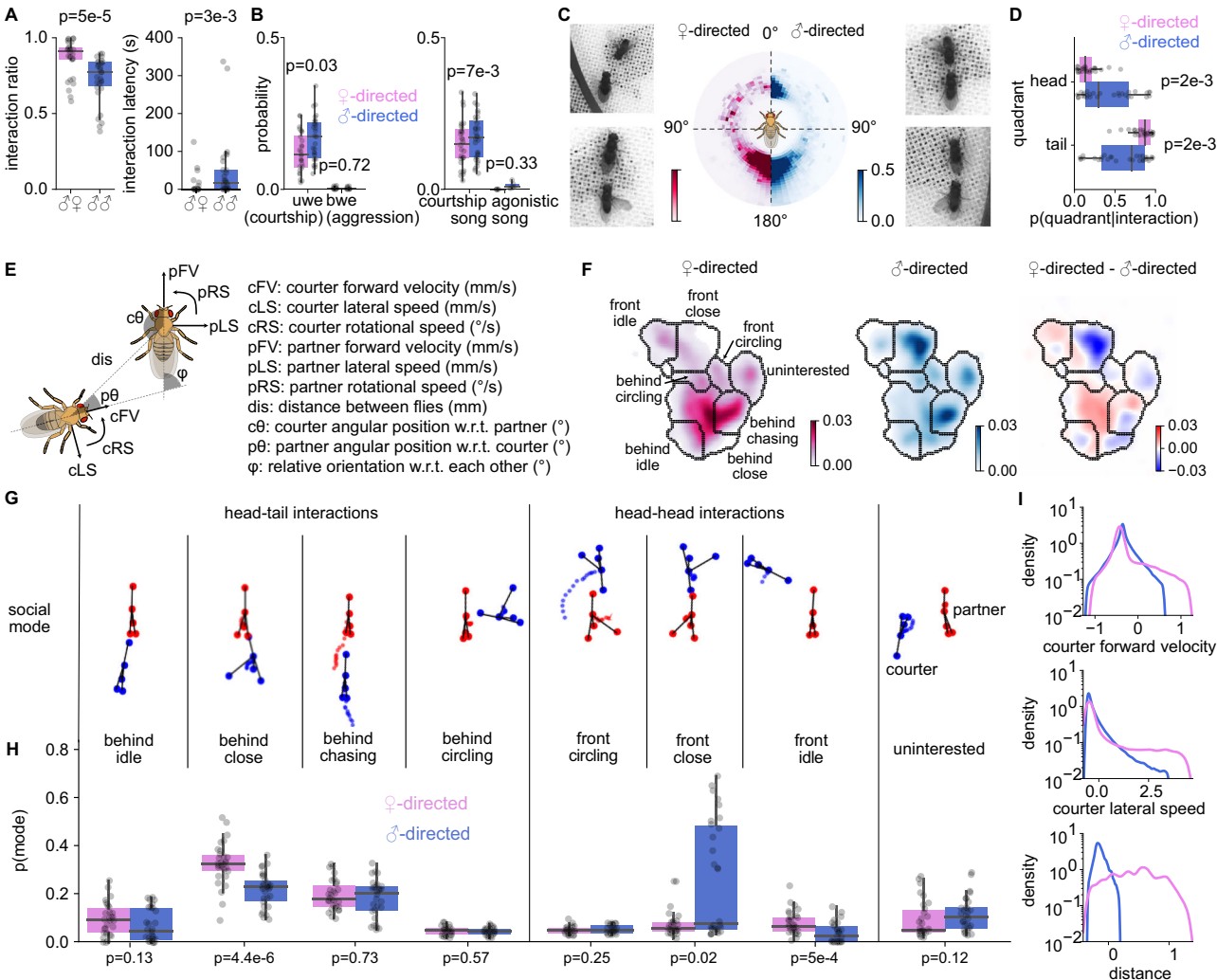

**Fig. 1 | Male- and female-directed courtship-like interactions exhibit different dynamics. A** Fraction of time spent interacting (left) and latency to interaction (right) for male-female (left, purple) and male-female pairs (right, blue). **B** Wing movements (left) and acoustic signals (right) produced in male-female (magenta) and male-male (blue) pairs. Unilateral wing extensions (uwes, left) indicate courtship-like interactions and typically lead to the production of courtship song (right). Bilateral wing extensions (bwes) indicate aggression and typically lead to agonistic song. **C** Position of a male counter around a female (left, magenta) or a male (right, blue) partner (see colorbar). 0° and 180° represent the partner's head and tail, respectively. Males are typically behind the female partner (tail interaction) or circle in front (arc) of her. By contrast, males spend more time directly in front of a male partner (head interaction). Males also tend to be positioned slightly farther from a male vs a female partner during tail interactions. Images show examples of tail (bottom) and head (top) interactions in male-female (left) and male-male (right) pairs **D** Ratio of time spent by the male counter near the head and tail of the partner

during interactions. **E** Parameters used to generate the social maps in I (see Methods for details). **F** Social maps for female-directed interactions (left), male-directed interactions (middle), and their difference (right). Color codes densities (left, middle) or their differences (right), see color bars. **G** Social modes represented by each cluster in the interaction state space (see Supplementary Videos 1–8). **H** The fraction of time spent in each social model during female-(magenta) and male-directed (blue) interactions. **I** Distribution of counter forward velocity, counter lateral speed, and distance when the male counter is singing near the head of the partner. The male is faster and at larger distances when singing near the head of a female partner and is slower and closer when singing near the head of a male partner. P-values were obtained using two-sided Mann–Whitney U tests. Dots in (**A**, **B**, **D**, **H**) show average for pairs of flies ($N = 30$ male-male and 30 male-female pairs). Boxplots in all figures: central mark indicates the median, the bottom and top edges of the box indicate the 25th and 75th percentiles, respectively. Whiskers extend to 1.5 times the interquartile range away from the box edges.

On a longer timescale (>50 ms), sine and pulse are sequenced into song bouts. We detected differences in song sequences only during head interactions (Figs. 2C–E, S4). During head interactions, males spent more time singing to a male than to a female partner (female: 10 ± 6% vs male: 22 ± 10%, Fig. 2C) and they sang more and longer sine song to a male than to a female (Fig. 2D, E). Additionally, males sang in different social modes near the head of a female and male partner (Figs. 2F, G, S5): Pulse-biased female-directed song occurred mainly during the "front circling" mode. Sine-biased male-directed song occurred mainly during the "front close" mode.

Together, these results show that while males use the same song signals when courting both female and male targets, they combine

these signals differently depending on the social context, leading to sex-specific song patterning. Thus, song patterning in flies exhibits a compositional structure, whereby complex behaviors arise from a limited set of behavioral primitives that are recombined according to context[62].

## Males use the same sensorimotor rules for male and female-directed singing

The sex-specific song patterning could arise from the males using sex-specific sensorimotor rules or from male and female partners providing differential feedback. To discriminate between these hypotheses, we first identified the song-patterning rules by combining a hidden

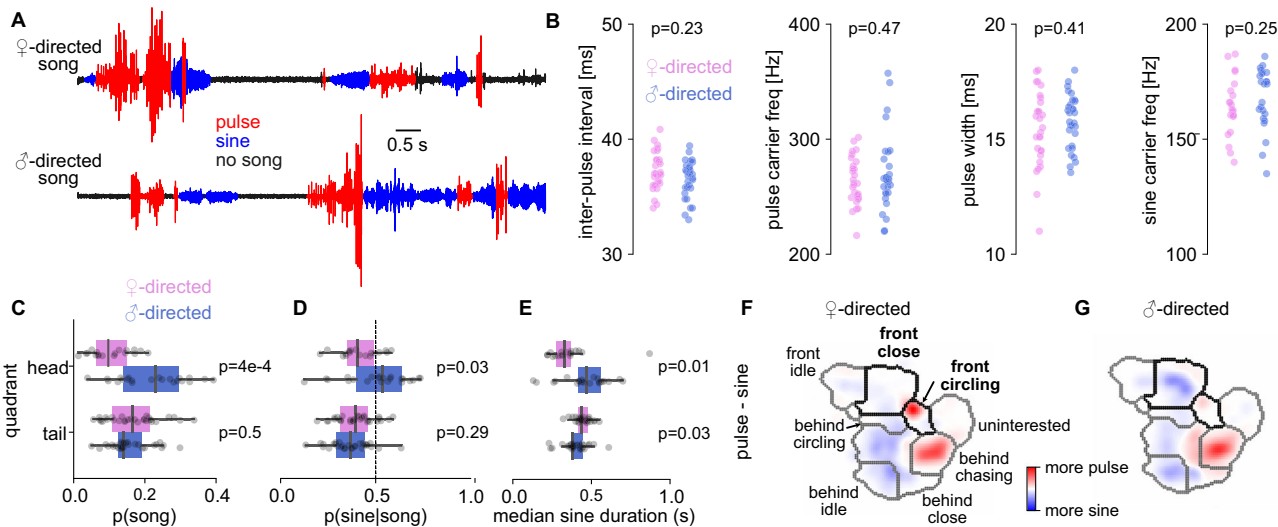

**Fig. 2 | Song patterns depend on the partner's sex during head interactions.**
**A** Traces of female (top) and male (bottom) directed courtship song. Both female- and male-directed song contain the pulse (red) and sine (blue) modes. **B** Song features on the short timescale (<50 ms) are identical in male (blue) and female-directed (blue) courtship song. **C**–**D** Probability of courtship song (**C**) and fraction of sine song out of all song (**D**) during head (top) and tail (bottom) interactions with a male (blue) and female (magenta) partner. Dashed line in D indicates an equal amount of pulse and sine. **E** Median duration of sine songs during head (top) and tail (bottom) interactions with a male (blue) and female (magenta) partner. **F**–**G** Difference between the social maps for pulse and sine song during female- (**F**) and male-directed (**G**) interactions showing sex-specific differences in song choice across social modes (see color bar). P-values were obtained using two-sided Mann–Whitney U tests. Dots in **B**–**E** correspond to average values for each NM91 fly pair. Each pixel in the social maps in **F**–**G** is averaged across fly pairs. *N* = 30 male-male and 30 male-female pairs.

Markov model (HMM) with generalized linear models (GLMs)[13,63–65] (Fig. 3A). In an HMM-GLM, each song patterning rule is represented by a GLM, which transforms partner feedback into the courting male's singing behavior: whether to produce pulses, sine songs, or nothing (Fig. 3B). The model's HMM component determines which of several rules is used during each time point. This combination of an HMM with GLMs can account for possible sex- and context-specific behavioral rules by allowing the model to flexibly switch cue-to-song mappings (Fig. 3C) according to the sex and behavior of the partner. As surrogate feedback cues, we used the kinematics of the courting male and the partner fly as well as their relative positions (Figs. 3B, 1E)[4,13].

When fitting the HMM-GLM to predict singing in our behavioral data, we found that males use the same set of rules for male- and female-directed singing: A model trained with combined data from male- and female-directed singing predicted song patterns as well as sex-specific models trained to explain singing towards only one of the two sexes. In addition, the sex-specific models generalized to explain singing to the sex not used for training (Fig. 3D). An HMM-GLM predicted song patterns better than an HMM or a GLM alone (Fig. 3C, E, F), implying that both the ability to switch rules (HMM) as well as the ability to flexibly respond to feedback cues (GLM) are crucial for explaining male singing. Models with three rules predicted the patterns of male- and female-directed song as accurately as a model with more rules (Fig. 3E). Additionally, the model with three rules predicted transitions between song modes better than models with fewer or more rules (Fig. 3F). The rules identified for male- and female-directed singing map to the rules previously identified for female-directed singing[13] (Figs. 3H–J, S6A): A non-interaction *whatever* rule, a pulse-biased *chasing* rule that is used when males are faster and the partner is farther, and a sine-biased *close* rule that is used when the male is slower and the partner is closer. Males use these rules with equal frequency towards either sex (Fig. 3G).

The same feedback cues determine singing for both partner sexes (Figs. 3K, S6B): The courting male's lateral speed and distance from the partner are the most important features for predicting singing towards male and female partners. By contrast, the cue that was most dependent on partner sex—the male's position around the partner (head or

tail, Fig. S6C)—is less important for song patterning. This indicates that the differences in song patterns do not arise simply from differences in the male's position around females and males, but from differences in his speed and distance from the partner during head interactions. Given that males use identical rules and cues for singing towards male and female partners, sex-specific song patterns likely arise from differences in partner feedback. We therefore examined and manipulated partner feedback.

## Partner feedback drives sex-specific song patterns

To characterize sex-specific partner responses to song, we constructed behavioral maps[58,59,66] of the partner's behavior during male singing (Fig. 4A). This revealed profound sex-specificity in the responses to courtship song: Female partners spend more time being idle and moving slowly or remaining stationary than male partners (Figs. 4B, S7). The female's idleness enables the male to initiate head interactions by rapidly circling towards her head while maintaining sufficient distance, likely to avoid startling her (Figs. 4C, D, S7C, D, Supplementary Video 9[60,67]). Since these head interactions are faster and occur at a distance, males use the chasing rule (Fig. 4E) and predominantly sing pulse song (Fig. 3H). By contrast, male partners typically run away or frequently sing back by extending their wing (Figs. 4A, B, S7). The male partner's constant movement prevents the singing male from circling in front. Instead, male-directed head interactions are initiated by the partner turning back to face and approach the singer (Fig. 4C, D, Supplementary Video 10). This slows the courting male and reduces the distance between both males, forcing the courter to choose the close rule (Fig. 4E) and sing more sine song (Fig. 3I). These observations suggest that partner feedback drives differences in rule use which in turn leads to sex-specific song patterns (Fig. 4F). Moreover, it implies that the head interactions between male partners and the subsequent increase in sine song are enforced by the male partner, not from deliberate choices by the courter.

To determine whether the causal factor is behavioral feedback rather than the partner's sex[67,68], we manipulated partner behavior by slowing male partners to resemble females, or making female partners turn back like males. We inactivated motor neurons to slow and stop

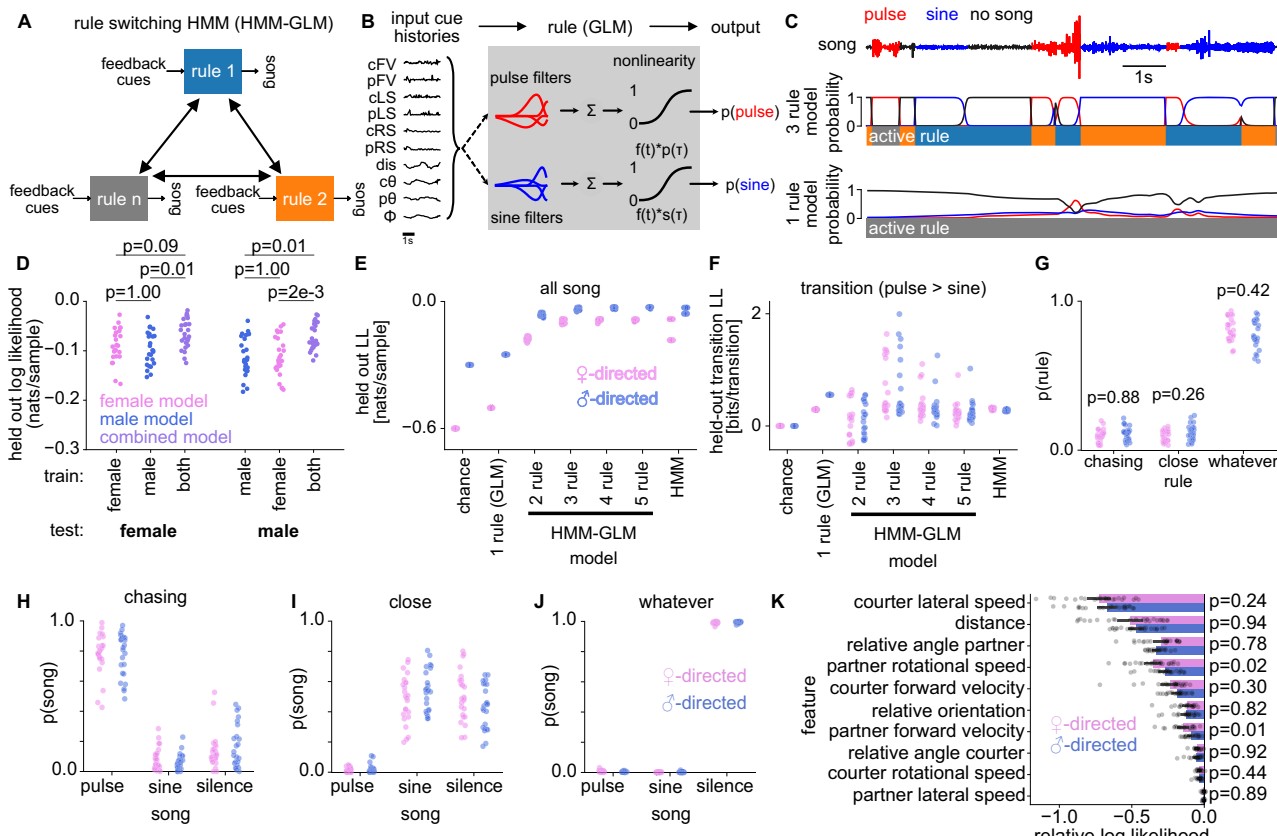

**Fig. 3 | Males use the same sensorimotor rules for male- and female-directed singing. A** Male- and female-directed singing were modeled using an HMM-GLM, combining a hidden Markov model (HMM) and a generalized linear model (GLM). Males switch between a fixed set of rules (boxes) that define sensorimotor trans-formations mapping feedback cues to song. **B** Each rule is implemented as a mul-tinomial GLM predicting song mode (pulse, sine, no song) from dynamical feedback cues extracted from tracking data (Fig. 1E). Each feature is filtered through linear filters for pulse and sine song. A nonlinear function determines song probabilities; no song equals one minus the probability of pulse and sine. **C** HMM-GLM with one (bottom) and three (middle) rules fitted to courtship song data. Shown is 10 s of interaction in which the male switches between pulse (red), sine (blue), and silence (black). The three-rule model accurately predicts song sequen-ces. **D** Goodness of fit (log-likelihood) for three-rule HMM-GLM models trained on male-directed (blue), female-directed (magenta), or combined (purple) data. Dots show likelihoods on held-out female (left) or male-directed (right) trials (N=25 each). All models generalize across interaction types. Models trained on combined

data perform slightly better. **E–F** Normalized log-likelihood on held-out female- and male-directed data for all song (**E**) and during song-mode transitions (**F**; pulse-to-sine transition shown). Dots represent fits with different initializations ($N = 20$). State-aware HMM-GLM models outperform GLM- or HMM-only models; models with more than three states provide no additional improvement. **G** Fraction of time males employ each rule during female- and male-directed interactions. Dots cor-respond to pairs ($N = 30$). **H–J** Probability of producing each song mode under the chasing (**H**), whatever (**I**), and close (**J**) rules toward males (blue) and females (magenta). Chasing predominantly produces pulse song, close mainly sine and silence, and whatever mostly silence. Dots correspond to pairs ($N = 30$ male-female and 30 male-male pairs). **K** Importance of each input feature for predicting song mode, computed as the reduction in model log-likelihood when the feature is randomly permuted. Dots represent individual training trials ($N = 24$ female-directed and 24 male-directed). All p-values were obtained using two-sided Mann–Whitney U tests.

one male in a pair using the light-gated anion channel GtACR1[69]. Inducing idleness in a male partner was sufficient to evoke frontal circling by the non-manipulated male (Figs. 4G, H, S8A) and eliminated sex-specific differences in rule use and song sequences (Fig. 4I, J). Next, we induced female turning by optogenetically activating pC1d neurons using the light-gated cation channel CsChrimson[70], which induces aggression and turning in females[71,72]. During pC1d activation, females turned back toward the courting males, leading to head interactions similar to those seen with male partners (Figs. 4K, L, S8B, C). The manipulated female partners were even closer to the counter than male partners, leading males to sing even more sine song using the close rule (Figs. 4M, N, S8D, E). Thus, sex-specific differences in song pat-terning are driven by behavioral feedback from the partner, not by their sex or by the courters' choice.

### Courtship song perception drives male-male head interactions
To test whether courtship song drives the turning of the male partner, we selectively disrupted song production or perception in male pairs.

We paired either a muted (wingcut) or a deaf (aristacut) with an intact partner capable of both singing and hearing. Disrupting acoustic communication strongly reduced head interactions (Fig. 5A, E right) and the few remaining head interactions were consistently initiated by the male capable of perceiving song.

In pairs containing a muted male, courtship by the muted male did not expose the intact male to song and accordingly courtship from the mute male predominantly led to tail interactions. In contrast, court-ship from the intact male exposed the muted male to song and led to head interactions, as the muted male turned toward the intact male in response to song (Fig. 5B–D). Similarly, in pairs containing a deaf male, courtship from the intact male toward the deaf male resulted mainly in tail interactions, as the deaf male was not able to perceive the intact male's song. Courtship from the deaf male exposed the intact male to song and led to head interactions induced by the intact male turning back in response to the deaf male's song (Fig. 5F–H).

Thus, perception of song is necessary for turning responses that generate head interactions. To test sufficiency, we exposed a pair of

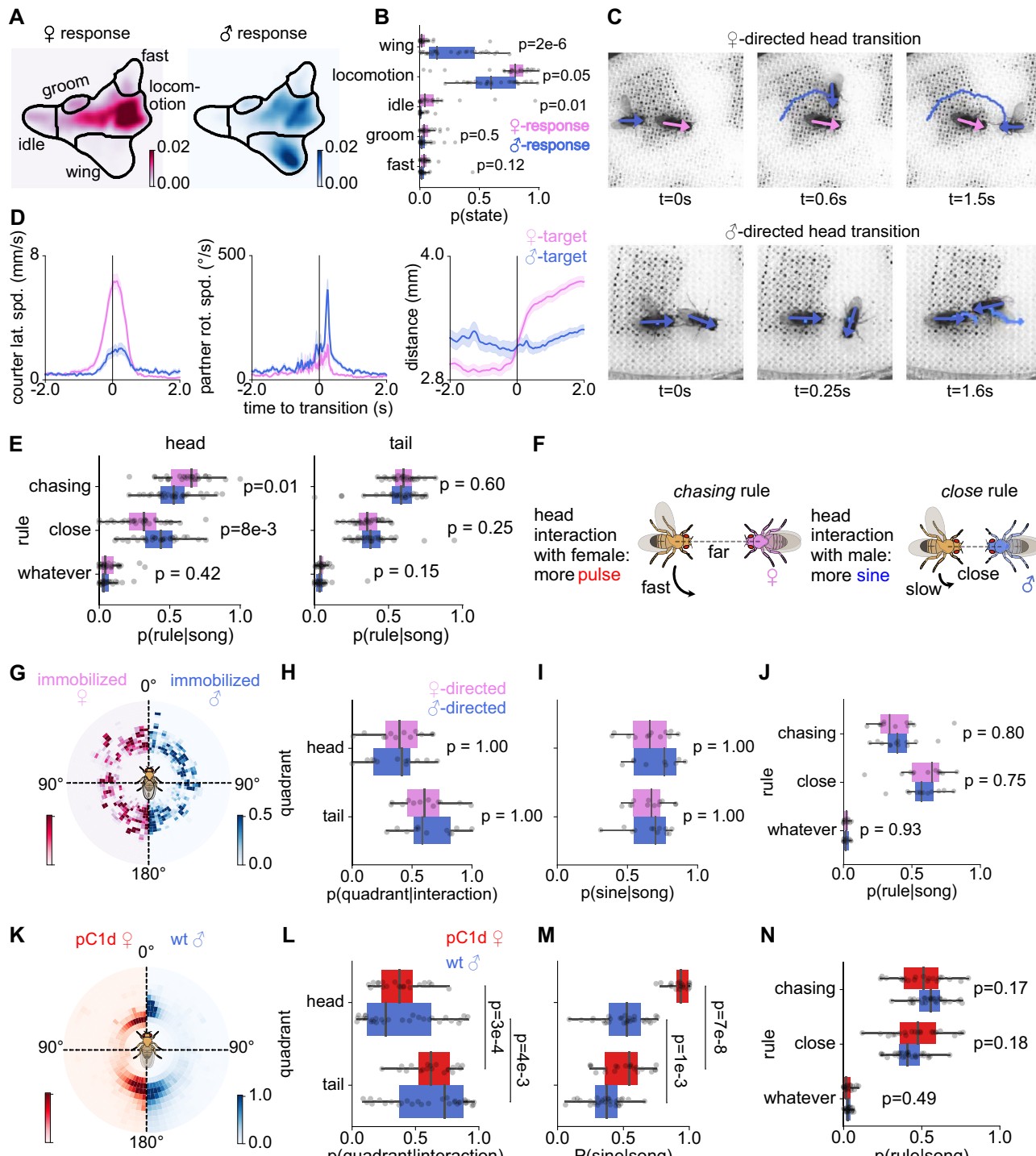

**Fig. 4 | Partner feedback drives sex-specific song patterns. A**, **B** Probability of behaviors of female (right) and male (left) partners while the counter male sings. Dots in **B** represent individual fly pairs ($N = 30$ male-female and 30 male-male pairs). **C** Transitions into head interactions differ by partner sex. With females (top), the counter moves to the partner's front. With males (bottom), the partner turns back toward the counter (see Supplementary Videos 9–10). **D** Counter lateral speed (left), partner rotational speed (middle), and distance (right) during transitions into head interactions with male (blue) or female (magenta) partners. Lines and shaded areas show mean ± s.e.m. ($N = 94$ transitions from 25 male-female pairs; $N = 88$ transitions from 26 male-male pairs). **E** Rule use when the counter sings near the head (left) or tail (right) of male (blue) or female (magenta) partners. Dots correspond to pairs ($N = 30$ male-female and 30 male-male pairs). **F** Sex-specific song sequences arise from differential rule use driven by partner feedback. Near the female head (left), males are faster and farther away and mainly produce pulse song. Near the male

head (right), males are slower, closer, and mainly produce sine song. **G** Position of a counter male around an immobilized female (magenta) or male (blue). **H** Fraction of time spent near the head and tail of immobilized partners. **I** Fraction of sine song (of all song) produced near the head and tail of immobilized partners. **J** Rule use when singing near immobilized partners. Dots in **H**–**J** represent fly pairs ($N = 12$ male-female and male-male pairs). **K** Position of a counter male around a pC1d-activated female compared with a wild-type male. **L** Fraction of time spent in head and tail quadrants around pC1d-activated females (red) and wild-type males (blue). **M** Fraction of sine song produced in head and tail quadrants. **N** Rule use when singing to pC1d-activated females versus wild-type males. Dots in **L**–**N** correspond to fly pairs ($N = 20$ pC1d females; $N = 30$ wild-type males). P-values were computed using two-sided Mann–Whitney U tests, except in **L**, **M**, where two-sided Wilcoxon signed-rank tests were used.

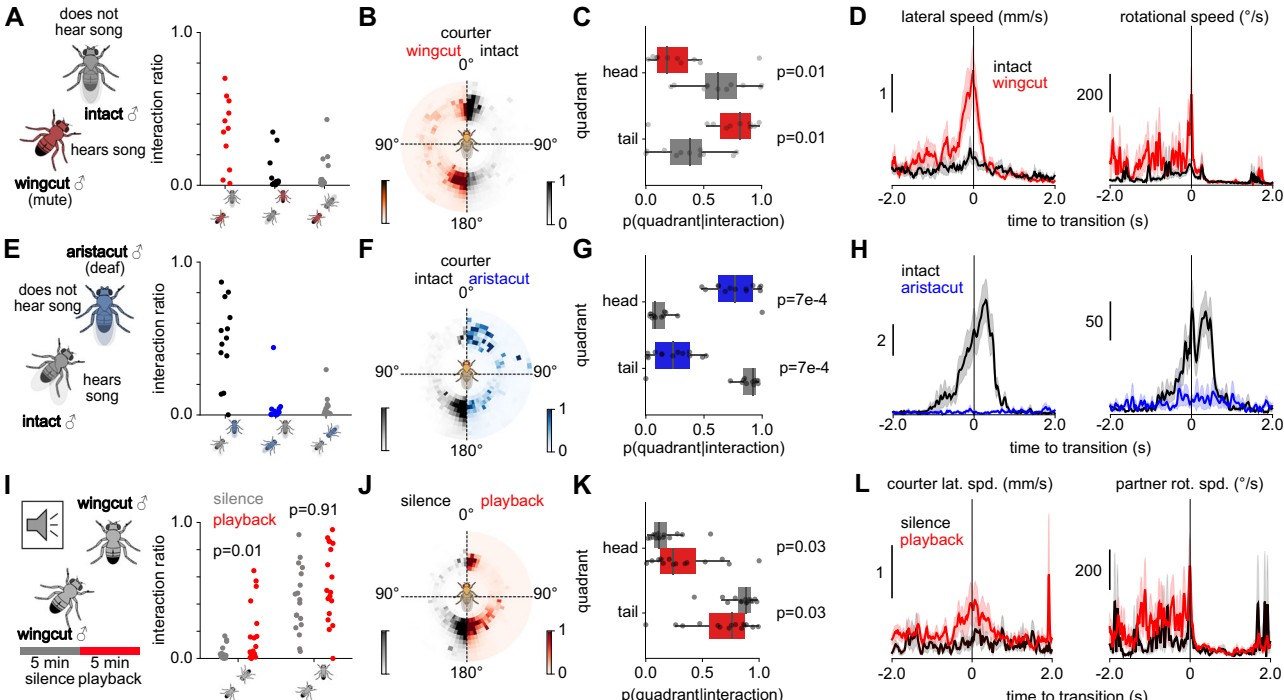

**Fig. 5 | Courtship song perception drives male-male head interactions.**
**A** Fraction of time spent in different configurations (mute male behind intact male, intact behind mute, and both facing each other) during interactions between an intact (gray) and wing-cut (mute, red) male (**A**–**D**, $N = 11$ pairs). **B** Position of the mute male when courting the intact male (red, left) and of the intact male when courting the mute male (black, right). **C** Fraction of interaction time spent by wing-cut and intact males in head and tail quadrants during courtship ($N = 11$ pairs). **D** Lateral (left) and rotational speed (right) of mute (red) and intact males (black) during transitions to head interactions (N = 24 mute-initiated and $N = 12$ intact-initiated transitions to head interactions). **E** Same as **A**, but for interactions between intact (gray) and aristacut (deaf, blue) males (**E**–**H**, $N = 13$ pairs). **F** Position of the intact male when courting the deaf male (black, left) and of the deaf male when courting the intact male (blue, right). **G** Same as **C**, but for interaction between deaf and intact flies ($N = 13$ pairs). **H** Same as **D**, but for interactions between intact (gray)

and aristacut (deaf, blue) males ($N = 9$ intact-initiated and $N = 2$ deaf-initiated transitions to head interactions). **I** Playback experiment with pairs of wing-cut (mute) males. Flies interacted for 5 min in silence followed by 5 min of continuous conspecific pulse-song playback. Right: fraction of time spent in different configurations during silence (black) and playback (red) ($N = 15$ pairs). **J** Position of the counter male during silence (black, left) and song playback (red, right). **K** Fraction of interaction time spent by the counter in head and tail quadrants during silence (black) and playback (red, $N = 15$ pairs). **L** Counter lateral speed (left) and partner rotational speed (right) during transitions to head interactions. Transitions occur primarily during playback and are initiated by the counter moving laterally to the front or by the partner turning back (silence: $N = 6$ transitions; playback: $N = 13$ transitions). p-values in **C**, **G** were computed using two-sided Mann–Whitney U tests; in **I**, **K** using two-sided Wilcoxon signed-rank tests. Lines and shaded areas show mean ± s.e.m.; dots represent fly pairs.

muted (wingcut) males to playback of recorded pulse song (Fig. 5I, left). Song playback elicited turning responses in the male being courted and increased head interactions (Fig. 5I–L). Together, these experiments demonstrate that perceiving song is both necessary and sufficient for head interactions.

On what timescales does the song affect turning? In females, song affects locomotion on two timescales: Immediately, as a trigger for slowing within milliseconds[31,73], and via integration over tens of seconds[74,75]. In males, only direct, immediate effects of song on locomotor behavior[31], aggression[44], or chaining[25,51] have been reported. Contrary to our expectation, song was not enriched immediately preceding male transitions from tail to head interactions (Fig. S9A). However, when evaluating song over longer timescales, we found more song in windows leading up to these transitions (Fig. S9B), indicating that turning was facilitated by song information integrated over tens of seconds. Thus, the mechanism underlying transitions from tail to head interactions is not simply phonotaxis[73] but likely involves a slow, song-driven buildup of the partner's social arousal. To further investigate the role of the partner's sexual arousal in inducing head interactions, we paired sexually satiated males with virgin mature males. The head interactions were rare in sexually satiated males with low sexual arousal compared to virgin males (Fig. S9C–E). These results illustrate that the perception of courtship song combines with a heightened internal arousal state to drive head interactions in males.

## Neural circuitry that links acoustic information with arousal and courtship induces turning

Central neurons that detect courtship song in males are well characterized: The auditory vPN1 and pC2l neurons detect pulse song and induce courtship behaviors[6,31,50] (Fig. 6A). Both neurons are thought to activate different subsets of pC1 neurons, which act as social command neurons that drive courtship or aggression[50,76–78]. To investigate the neural basis of song-induced male turning behavior, we tested the role of auditory vPN1 and pC2l neurons in driving head interactions and investigated which subsets of pC1 neurons drive these interactions.

Male-specific vPN1 neurons are necessary for song-induced male chaining[50]. Chaining is only observed in groups of males and results in a chain of several males orienting head-to-tail[49,51]. We tested whether the vPN1 neurons also induce head interactions between males. Activation of vPN1 in male pairs failed to induce head interactions and instead strongly increased tail interactions (Fig. 6B–E). Next, we tested the role of the pC2l neurons, which detect pulse song and induce acceleration and singing in solitary males[6,31]. Unlike vPN1, optogenetic activation of pC2l neurons induced head interactions in male pairs (Fig. 6F–I, by inducing a turning response of the partner male during activation (Fig. 6J). To show that pC2 neurons contribute to head interactions driven by turning responses, we silenced pC2 neurons using tetanus toxin (TNT) (Fig. 6K). A pC2-silenced or control (TNT-inactive) male was paired with a P1a-activated male to induce high

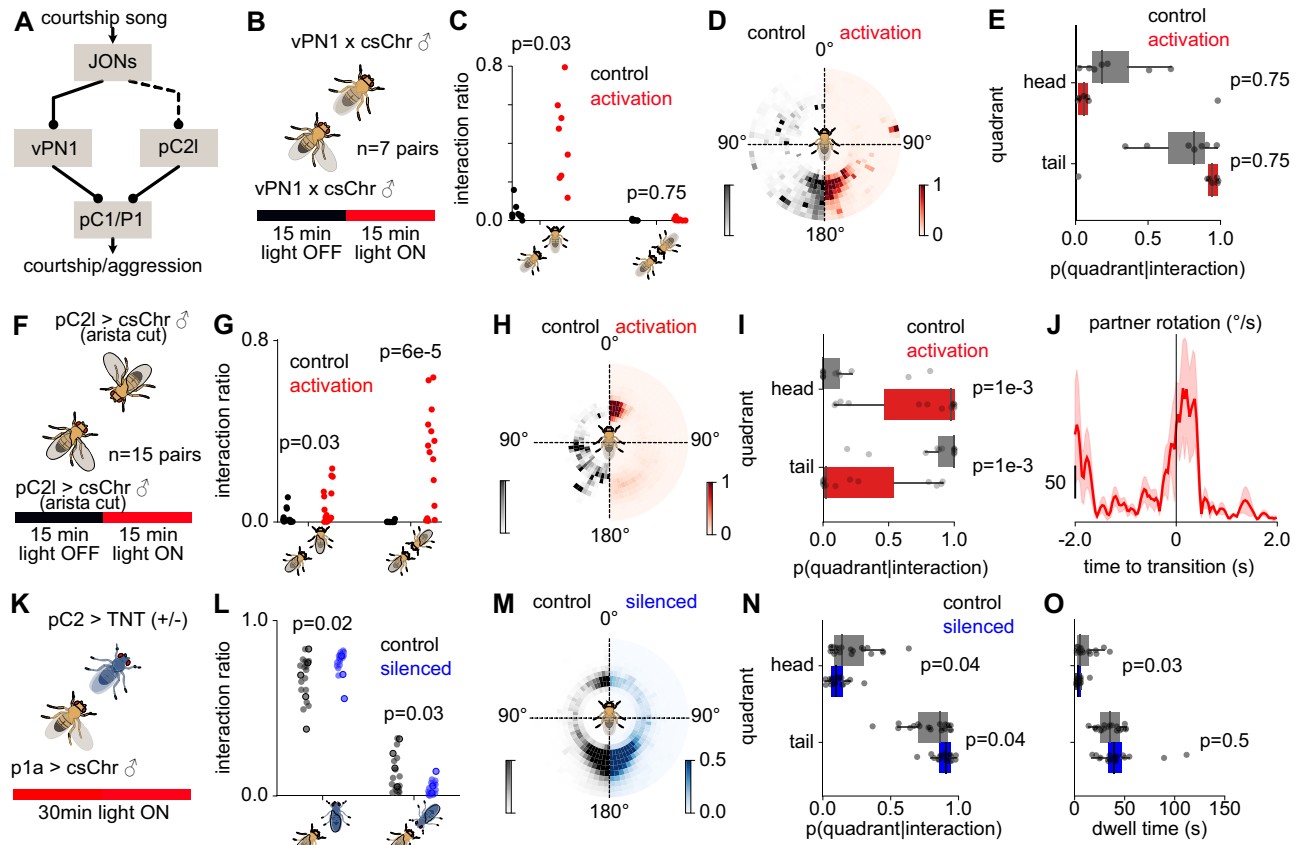

**Fig. 6 | Song detection neurons drive head interactions. A** Song detection pathways in the fly brain. **B** Optogenetic activation of vPN1 neurons (**B–E**, $N = 7$ pairs). Two males that expressed red-shifted channel rhodopsin csChrimson in vPN1 neurons are paired for 30 min in the assay. During the first 15 min, the red LED was OFF (control) followed by vPN1 activation for 15 min (red LED ON). **C** Fraction of trial time spent in tail (left) and head interactions (right) during control (black) and vPN1 activation (red). **D** Position of the counter male around the partner during control and vPN1 activation. **E** Fraction of interaction time spent in head (top) and tail (bottom) quadrants during control (black) and vPN1 activation (red). **F–I** Same as **B–E**, but for activation of pC2l neurons ($N = 15$ pairs). **J** Partner rotational speed during the transition to head interactions. Head interactions are driven by the partner male turning back to the counter during pC2l activation (increase in partner rotation) (control: N=0 transitions, activation: $N = 4$ transitions from 4 pairs). **K** pC2 silencing using TNT. To increase male-male interactions, pC2-silenced males

were paired with P1a-activated males (csChrimson in P1a neurons) and exposed to red light for the whole duration of the experiment. **L** Fraction of trial time spent by the P1a-activated male in head and tail interactions with pC2-silenced (TNT, blue) and control (TNT-inactive, black) males. **M** Position of the P1a-activated male around pC2-silenced (blue, right) and control (black, left) males. **N** Fraction of interaction time spent by P1a-activated male in head and tail quadrants of pC2-silenced (blue) and control (black) males. **O** Dwell times of the P1a-activated male in the head and tail quadrants of pC2-silenced (blue) and control (black) males. Dots in **L**, **N**, **O** correspond to averages for each fly pair (control: $N = 21$ pairs, silencing: $N = 18$ pairs). The p-values in **C**, **E**, **G**, **I**, are computed using two-sided Wilcoxon signed rank tests, in **L**, **N**, **O** using two-sided Mann–Whitney-U tests. Lines and shaded areas show the mean $\pm$ s.e.m. across all transitions. Dots correspond to averages for each fly pair.

levels of male-male interactions. The pC2-silenced males initiated fewer head interactions and turned less towards the P1a-activated courter compared with control males (Fig. 6L–N). In addition, head interactions were shorter in pC2-silenced males (Fig. 6O).

Together, these results show that, distinct auditory pathways drive specific responses to song in males that lead to different modes of male-male interactions, with vPN1 driving tail interactions (Fig. 6B–E[50]), and pC2l driving head interactions (Fig. 6F–J). In females, the pC2l neurons have been shown to induce slowing down[31]. Thus, the sexually dimorphic behavioral feedback to courtship song (males turning back and females slowing down) is mediated by the shared song detector neurons, pC2l[31,79].

The pC2l neurons drive song in males via the descending pIP10 neurons[80]. Recent studies suggest that pC2l neurons also connect to P1a neurons, a male-specific subset of the pC1 neurons, which accumulates social cues and encodes a persistent social arousal state[6,81]. Another pC1 subset, called pC1x, is necessary for song-induced male-male aggression[44]. To test which of the pC1 subsets, P1a or pC1x, drive head interactions and courtship in male pairs, we activated the pC1

neurons or their subsets P1a and pC1x. Optogenetic activation of all pC1 neurons in pairs of males (Fig. 7A), each deafened to remove effects from endogenous song, promoted both tail and head interactions (Fig. 7B, C). Activation of the pC1x subset induced more tail interactions (head: $10 \pm 6\%$ vs. tail: $90 \pm 6\%$, $p = 2e-3$, Fig. 7K–N) and P1a activation (Fig. 7F, G) induced more head interactions (head: $68 \pm 24\%$ vs. tail: $32 \pm 24\%$, $p = 0.01$, Fig. 7H–J), suggesting a specific role of P1a neurons in inducing head interactions in male pairs. Overall, the optogenetic activation data point to the existence of two pathways that link hearing song in male partners to specific feedback behaviors: First, one in which song is detected in vPN1 neurons to drive chaining, possibly via the pC1x neurons. And second, song detected by the pC2l neurons increases the male's arousal and drives turning and singing towards other males via the P1a subset (Fig. 7O).

## Discussion

Here, we have shown how the interplay between external behavioral feedback and internal sensorimotor processing shapes flexible social behavior by comparing courtship-like interactions directed at male

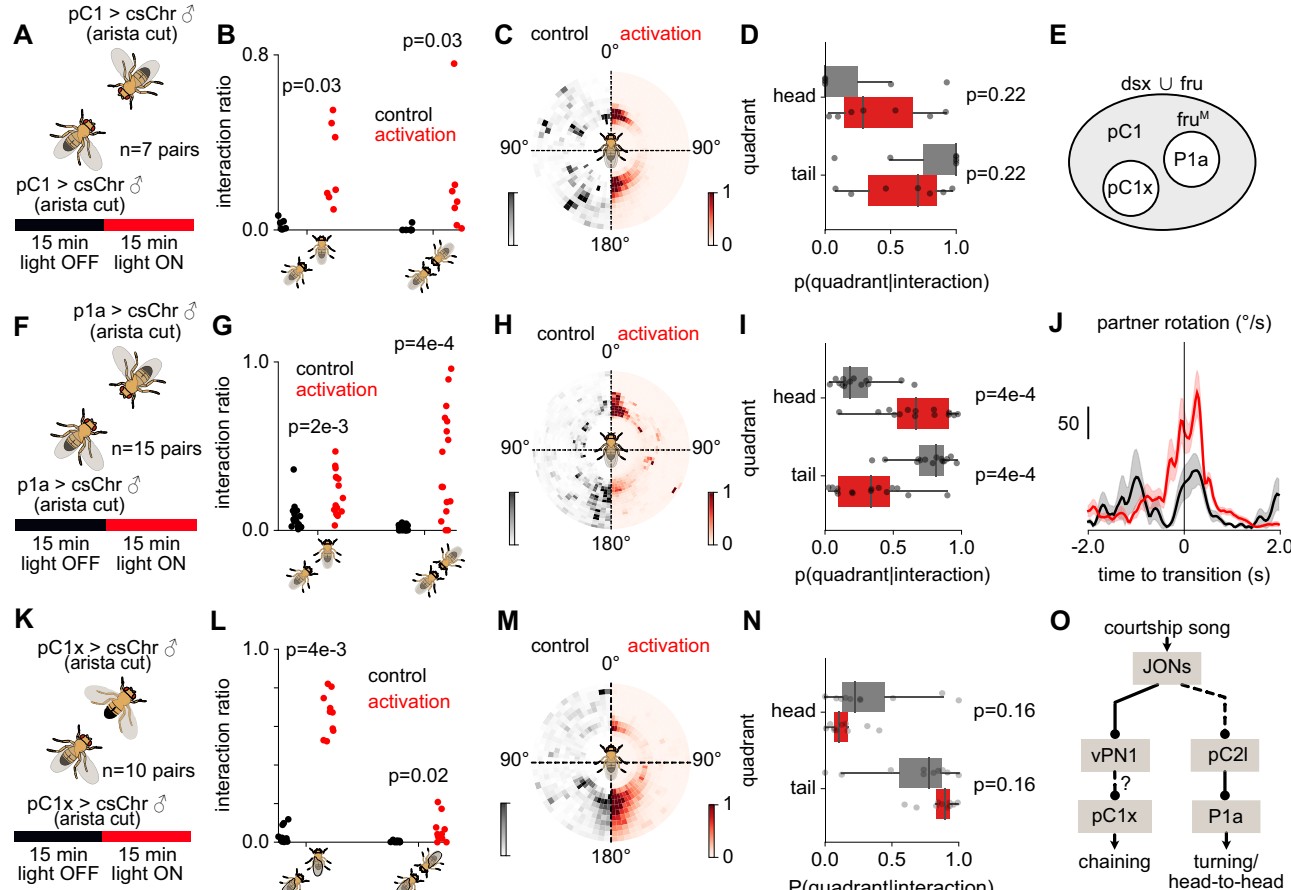

**Fig. 7 | Song induces head interactions through a specific neural pathway.**
**A** Experimental procedure for optogenetic activation of pC1 neurons ($N = 7$ pairs). Two males that expressed csChrimson in pC1 neurons are paired for 30 minutes in the assay. For the first 15 minutes of the trial, the red LED was OFF (control) followed by pC1 activation for 15 min by turning ON the red LED. **B** Fraction of trial time spent in head (left) and tail interactions (right) during control (black) and pC1 activation (red). **C** Position of the courter male around the partner during control and activation of pC1 neurons. **D** Fraction of interaction time spent in head (top) and tail (bottom) quadrants during control (black) and pC1 activation (red). Dots in **C, E** correspond to averages for each fly pair ($N = 7$ pairs). **E** pC1 represents a broad set of neurons in the central brain of both females and males and consists of multiple subtypes expressing sex-determination genes doublesex (Dsx) and fruitless (Fru). A specific subset of neurons known as P1 expresses FruM proteins and is

present only in males. **F–I** Same as **B–E** but for activation of P1a neurons. Dots in **G** and **I** correspond to each fly pair ($N = 15$ pairs). **J** Partner rotational speed during the transition to head interactions during P1a activation experiments. Head interactions are driven by the partner male turning back to the courter during P1a activation (increase in partner rotation). Solid lines correspond to mean and shaded area s.e.m. across all transitions (control: $N = 4$ transitions from 4 pairs, activation: $N = 28$ transitions from 10 pairs). **K–N** Same as **B–E** but for activation of pC1x (pC1SS2) neurons. Dots in **L, N** correspond to each fly pair ($N = 10$ pairs). **O** Putative circuit for song-induced feedback behaviors in male fruit flies. Distinct pathways drive chaining and head interactions, respectively, the former through the vPN1 neurons and a non-P1a pC1 subtype such as pC1x, and the latter through the pC2l neurons providing excitatory inputs to P1a. P-values are computed using two-sided Wilcoxon signed rank tests.

and female partners. We found that the dynamics of courtship interactions change with the sex of the partner such that courting males were often facing the head of another male close up whereas often facing the tail of females (Fig. 1). This resulted in differences in the sequences of male- and female-directed courtship song with male partners receiving more sine song and females more pulses (Fig. 2). We then showed that courters use a fixed set of three sensorimotor rules for sequencing male- and female-directed song (Fig. 3). The differences in song sequences are driven by sex-specific partner feedback, which in turn drives sex-specific rule use by courters (Fig. 4). Sex-specific partner feedback is mediated by distinct neural pathways that link hearing song (Fig. 5) with behavior (Figs. 6, 7). Together, these findings show that differential social feedback by male and female partners can elicit differential actions through the same sensorimotor processing rules, leading to flexible context-dependent behavioral patterns.

We (Fig. 3) and others[13] have shown that *Drosophila* males pattern their courtship song using three sensorimotor rules, but how these rules are represented in the brain is unclear. Previous studies in non-

human primates suggest that rule encoding in the brain manifests as correlated activity between sensory and action-related regions[82,83]. Transitions between rules can alter these correlations through changes in sensory-motor cortical connections or through shared inputs from rule-encoding neurons to sensory and motor areas[82]. In rodents as well as human and non-human primates, neurons in decision-making regions such as the prefrontal cortex, orbitofrontal cortex (OFC), and striatum encode rule identity via changes in neuronal firing rates[5,84–88]. Our findings on differential rule use in courting males according to partner behavior in *Drosophila* lay the groundwork for identifying the neural circuits that encode contexts and select rules in an experimentally accessible model organism with a known connectome[89].

In *Drosophila* males, the context-dependent rule use (Fig. 3) is driven not by the partner's sexual identity but by their feedback (Fig. 4). Rule selection is therefore likely mediated by ascending and visual neurons that encode the locomotor state and the distance of the male courter from his interaction partner (Figs. 3M, S6B). Self-motion signals could be conveyed via central re-afferences[90] or from peripheral motor centers and proprioceptors[91–93]. High-order visual features

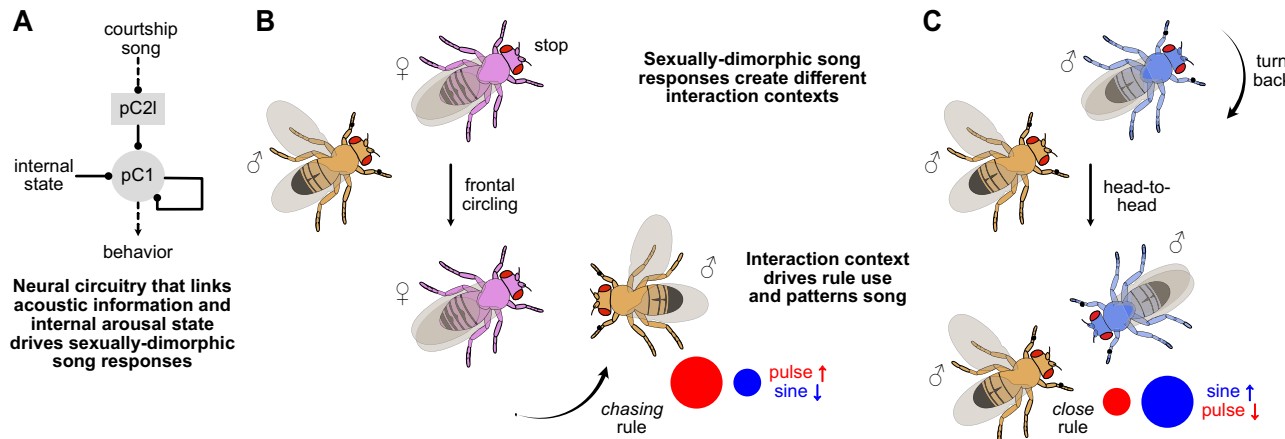

**Fig. 8 | Sexually dimorphic behavioral responses to courtship song drive sex-specific rule use. A** Courtship song is processed by the sexually dimorphic pC2l neurons[31] and pC1 neurons[50,104]. Sex-shared pulse song detector neurons pC2l drive sexually dimorphic behavioral responses: Slowing down or stopping in females and turning back in males. **B** Song from a courting male induces sexual receptivity in the female partner and slows her, prompting the male to circle to her front. The counter predominantly uses the chasing rule to sing in this context. **C** On the other hand, song may induce sexual arousal in a male partner via pC2l and P1a neurons. He turns back and sings to the counter, establishing head interactions between males. The counter predominantly uses the close rule to sing in this context.

that encode the distance and movement of the partner are processed in the lobular columnar neurons[94–96]. This context information is then likely integrated in the superior medial protocerebrum, likely involving subsets of pC2 (Fig. 6F–O) and pC1 neurons (Fig. 7A–J)[6,13,31,81]. Contributions of other integrative centers, such as the mushroom bodies or the central complex, cannot be ruled out[97]. All these circuits link sensory cues to motor outputs through descending neurons that target pre-motor centers in the ventral nerve cord[93,98–100]. This is supported by recent studies showing that song patterns are shaped by differential recruitment of at least three types of descending neurons: pIP10, pMP2, and a descending neuron that is part of a disinhibitory circuit downstream of P1a[6,13,61].

The behavioral rules that specify how sensory cues are transformed into song can generate robust male- and female-directed song despite large differences in interaction dynamics, because the rules are sensitive to some, and invariant to other sensory cues: Song patterns are not shaped by the male position at the head or tail of the partner, but by differences in the partner's speed and distance (Figs. 3, 4, S6). This implies that the male's sensory experience during courtship can be split into two subspaces: an output-potent subspace (e.g., speed, distance) that drives singing, and an output-null subspace (e.g., angular position) that does not[101,102]. This division enables males to generalize their singing to novel contexts, such as head interactions with other males. The concept of the output-null space originates in motor neuroscience, where it describes how neuronal activity in the primary motor cortex of non-human primates influences muscle activity during the execution, but not the preparation, of movement[101]. Preparatory neuronal activity resides in the output-null subspace, allowing planning without affecting execution. Similarly, in male singing, positional changes around the female[60] do not alter the song pattern because male position lies in the output-null subspace. This enables males to reposition to the female's rear in preparation for copulation without disrupting their ongoing song pattern.

In our assays, a defining feature of male-male courtship-like behavior in *Drosophila* is the head interaction, initiated when the male partner turns back toward the courter. Both courtship song perception and heightened arousal are required to induce the partner's turning (Fig. 5). Information about song in the partner is relayed via distinct auditory pathways with opposing effects: vPN1 drives tail chasing (Fig. 6C–F) and chaining[50], while pC2l drives head interactions (Fig. 6F–O). Silencing of pC2 neurons did not completely abolish the head interaction between males (Fig. 6L–N), suggesting that additional

song detection neurons such as pC1[50,103] can induce head interactions. Our findings indicate that vPN1 and pC2l effect distinct behavioral responses via connections to different subpopulations of pC1 neurons: Effects of activation of the pC1x subset resemble those of vPN1 activation (Figs. 6D, E and 7M, N), suggesting a functional connection between both neurons, although this remains to be tested. By contrast, pC2l is thought to be connected to the P1a subset[6,81], and indeed activation of both cell types elicits head interactions (Figs. 6G–I, 7G–I). The P1a neurons control social arousal[76] and are part of an integrator network that might slowly accumulate song inputs from pC2l (Fig. S9A, B). Once the male's arousal level is sufficiently high, the song-induced activity in P1a neurons surpasses inhibitory signals from male pheromonal cues[36,104,105], leading to disinhibition of the partner male's courtship circuitry and male-directed singing. Once established, head interactions are likely stabilized internally by recurrent circuitry downstream of P1a[77] but likely also through a positive feedback loop formed by P1a driving song in one male, which activates P1a and drives singing in the other male. Further experimental validation is required to confirm the proposed neural circuit and song integration mechanism underlying male-male head interactions.

Overall, our findings demonstrate that sex-specific song patterning in *Drosophila* males emerges from context-dependent application of sensorimotor rules driven by sexually-dimorphic partner feedback. For example, female receptivity induced by male song slows her movement, triggering frontal circling where the male engages the *chasing* rule that elevates pulse song. On the other hand, male arousal induced by courtship song triggers a head encounter where the *close* rule is engaged, dominated by sine song (Fig. 8). The sexually-dimorphic song responses are likely mediated through sexually dimorphic song detection neurons in the pC2l and pC1 clusters (P1a in males and pC1a/c in females)[31,72,79]. Although learning based adaptations to sensorimotor processing cannot be excluded, plasticity likely plays only a minor role under our conditions, since we used sex naïve flies. Instead, males flexibly recombine existing rules—exhibiting compositionality—a resource-efficient strategy for generating diverse behavioral sequences from a limited repertoire without the need for learning new actions[106–108]. While compositionality has been demonstrated in biological and artificial neural networks for cognitive decision-making tasks[106–109], our study extends it to natural behaviors. Our results further show that in flies, compositionality exists not only at the level of rules (input-output mapping), but also in the sequencing of motor outputs[62,110,111]. The sex-specific song sequences towards

females and males do not arise from differences in the characteristics of the song signal itself (carrier frequency, width, duration, IPI), but from males patterning the same signals in different combinations towards the two sexes, depending on the interaction context (Fig. 2). More broadly, our findings highlight how flexible, context-dependent behaviors can emerge from a static rule set and limited behavioral repertoire via feedback-driven modulation, proposing a mechanism for flexibility of innate behaviors[2,7–10,112,113]. Remapping existing rules to novel behavioral outputs likely also underpins the evolutionary diversification of behavioral repertoires[114,115].

## Methods

### Experimental animals

Flies were raised on a standard cornmeal-agar medium at 25 °C with 60% humidity and a 12-h light/dark cycle. Within six hours of eclosion, virgin flies were separated by sex and housed in groups of 10–15 flies. For the experiments, we used flies that were between three to seven days old. For optogenetics, flies were housed on retinal food after eclosion. The retinal food was prepared by supplementing standard fly food with 400 μM all-trans-retinal (Sigma-Aldrich R2500) dissolved in 99% ethanol. To prevent optogenetic activation, light exposure was minimized by wrapping the vials in aluminum foil. The fly lines used in this study are listed in Table 1.

### Behavioral assays

The behavior chamber was circular with a 44 mm diameter and 1.9 mm height unless otherwise specified. The large diameter allowed a wide range of fly behaviors and interactions whereas the narrow height prevented the flies from lifting off. The chamber and the lid were made of transparent acrylic to allow video recording. The lid was movable and contained a small opening for loading the flies. The chamber's walls were sloped to prevent the flies from climbing to the ceiling. The floor of the chamber was tiled with 16 microphones (Knowles NR-23158) embedded in a custom PCB (Modified from ref. 53). The acoustic signals were amplified using a custom-build amplifier (Modified from ref. 53) and digitized using a data acquisition card (National Instruments PCIe-6343) at a sampling rate of 10 KHz. The microphones were covered with two layers of thin nylon mesh (50 mm diameter) for the flies to walk on. Fly behavior was recorded with a USB camera (FLIR flea3 FL3-U3-13Y3M-C, 100 frames per second (fps), 912 ×920 pixels) equipped with a 35 mm f1.4 objective (Thorlabs MVL35M1) mounted above the chamber. The chamber was illuminated using a circular array of white LEDs placed between the camera and the chamber. For optogenetic experiments, the chamber was illuminated with blue LEDs (470 nm, 1.5 μW/mm$^2$) whose wavelengths do not interfere with red and green wavelengths used for optogenetic manipulations. Song was played back using a loudspeaker (Hi-Vi Research® Loudspeaker Model F6) placed on one side of the chamber.

Before the first experiment of each day, the chamber was prepared in two steps. First, the chamber and its lid were cleaned with ethanol, rinsed with water, and allowed to dry. The lid was then coated with Sigmacote (Sigma-Aldrich) to prevent flies from walking on the chamber ceiling. Second, to promote interactions between flies during the experiment, the chamber was perfumed with male and female pheromones by loading it with ∼5 male and ∼5 female flies for 5 min. For the experiments, flies were loaded individually into the chamber using an aspirator. Since the flies are most active at dawn, we conducted all experiments within 3 h of the behavioral incubator lights switching on. Recordings were terminated when copulation occurred or after 15 minutes unless otherwise specified. Synchronized recordings of audio, video, and delivery of optogenetic stimuli were controlled using custom software (https://janclemenslab.org/etho).

We took several steps to promote courtship behavior in male-male pairs: First, we did not add a resource such as food or a decapitated or freeze-killed female, which typically drives aggression[43–45]. Second, since socially isolated flies are more aggressive, we group-housed males before the experiments[116]. Third, we perfumed the behavioral chamber with male and female pheromones by placing 5 males and 5 females into the chamber before each set of experiments[4]. Although perfuming may reduce the impact of volatile sex pheromones during courtship, interactions will still be influenced by the detection of non-volatile sex pheromones on the partner's cuticle[117–119]. Lastly, we did not use genetic manipulations that interfered with the males' ability to discriminate sex, since we wanted to study naturalistic male-directed singing. Instead, we used NM91, a strain of *Drosophila melanogaster* with a high motivation to court females[4].

### Optogenetic experiments

For the optogenetic experiments, flies were collected and placed on retinal food immediately after eclosion. For neural activation using csChrimson[70] or inactivation using GtACR1[69], the chamber was illuminated using four red (625 nm) or green (525 nm) LEDs, respectively. A 500 nm short-pass filter (Edmund Optics, 500 nm 50 mm diameter, OD 4.0 Shortpass Filter) was attached to the camera's objective to prevent the optogenetic stimulation from interfering with video recording.

## Table 1 | Fly lines used in this study

| Figure | Name | Genotype | Source | Donor / Identifier |
|---|---|---|---|---|
| 1, 2, 3, 4A–E, 5 | wild-type | *Drosophila melanogaster* NM91 | 4 | Peter Andolfatto |
| S1C, D | wild-type | *Drosophila melanogaster* oregonR | | |
| 4G–J | vGlut | ;VGlut1[OK371]-GAL4/+; UAS-GtACR1.d.EYFP(attP2)/+ | 69 | Martin Goepfert |
| 4K–N | pC1dSS1 | ; R35C10-p65.ADJK73A/+; 20xUAS-IVS-CsChrimson.mVenus(attP2)/R71A09-GAL4.DBD(attP2) | 71 | SS56987 |
| 6C–E | vPN1 | ;R72E10-p65.AD/+; 20xUAS-IVS-CsChrimson.mVenus(attP2)/VT009665-GAL4.DBD | 50 | Adrián Palacios Muñoz, BL #71127 and #74285 |
| 6G–J | pC2l | ;UAS(FRT.STOP)CsChrimson.mVenus(attp14)/+; GMR42B01-Gal4(attP2)/8xLexAop2-FLP(attp2),dsx-LexA | 31 | Vivek Jayaraman |
| | pC2-TNT(+) | ;+/+; GMR42B01-Gal4(attP2)/uAS-TNT | | Anne von Philipsborn, |
| 6L–O | pC2-TNT(-) | ;+/+; GMR42B01-Gal4(attP2)/UAS-TeTxLC.(-)V | | BL #69895 and #28441 |
| 7B–D | pC1 | ;GMR71G01-GAL4(attP40)/+;20xUAS-IVS-CsChrimson.mVenus(attP2)/+ | | BL #39599 |
| 7G–J | P1a | ;R15A01-p65.AD(attP40)/+; 20xUAS-IVS-CsChrimson.mVenus(attP2)/R71G01-GAL4.DBD(attP2) | 76 | David Anderson |
| 7L–N | pC1x (pC1SS2) | ;+/+;20xUAS-IVS-CsChrimson.mVenus(attP2)/VT002064-p65.ADZp(attP2), dsxZpGal4.DBD | 44,121,122 | BL #86836 or SS59911 |

**Immobilization experiments.** Partners were immobilized by using GtACR1 to inhibit all glutamatergic neurons, which include the motorneurons in the fly. To promote courtship towards the immobilized partners, we activated the P1a in the counter flies using csChrimson prior to the immobilization. The pair was initially exposed to red light for four minutes to induce a persistent arousal state in the counter by activating P1a neurons. Then, the pair was exposed to green light, to immobilize the male or female partners by inhibiting their motorneurons.

**PC1d activation experiments.** Wildtype NM91 males were paired with female flies expressing CsChrimson in pC1d neurons. Each experiment lasted 14 min, with the red light being OFF during the first 7 minutes (control) and ON during the next seven minutes (experiment).

**pC2l and vPN1 activation.** Two male flies that expressed uAS-csChrimson in vPN1[50] (Fig. 6B–E) or pC2l[31] (Fig. 6F–J) neurons were paired for 30 min. The flies interacted for the first 15 min of the trial under normal light (control). Red light (27 μW/mm²) was turned ON for the next 15 min (experiment) which activated the vPN1 or pC2l neurons in the flies. Pilot experiments with pC2 activation in our large behavioral chamber (diameter 42 mm) showed that the flies frequently sang to the walls during activation. To promote interactions, we subsequently used a smaller chamber with a diameter of 16 mm and height of 3.25 mm.

**pC2 silencing.** Males expressing tetanus toxin light chain, uAS-TNT[120], to block synaptic transmission in pC2 (R42B01) neurons were paired with males expressing uAS-csChrimson in P1a[76] neurons for 30 min (Fig. 6K). For controls, males expressing uAS-TNT inactive in pC2 (R42B01) neurons were paired with P1a-csChrimson males. A red light (27 μW/mm²) was turned ON for the whole duration of the trial to arouse the P1a-csChrimson male to interact with high intensity towards both control and pC2-silenced males. All trials with an interaction ratio less than 70% were excluded from analysis. This provided a common baseline for comparing the behavioral responses of pC2-silenced and control males.

**pC1, pC1x and P1a activation.** Two deaf male flies that expressed uAS-csChrimson in pC1[50] (Fig. 7A–D), pC1x[44,121,122] or P1a[76] (Fig. 7F–N) were paired for 30 minutes. The flies interacted for the first 15 min of the trial under normal light (control). Red light (27 μW/mm²) was turned ON for the next 15 min which activated the pC1, pC1x, or P1a neurons (experiment). The flies were deafened by cutting their arista under cold anesthesia at least 12 h before the experiment.

**Auditory manipulations.** For experiments in Fig. 5A–D, we paired a mute (wing-cut) male with an intact male (both NM91). For experiments in Fig. 5E–H, a deaf male was paired with an intact male (both NM91). For experiments in Fig. 5I–L, we used two mute (wing-cut) males (NM91). The flies were paired for 10 min, of which they interacted for the first 5 min in silence control. After 5 min, an artificial pulse song with a pulse width of 4 ms and an inter-pulse interval (IPI) of 36 ms was played back continuously for the next 5 min (playback experiment). For song playback, we used the protocol by ref. 123. We played back ten sound amplitudes at increasing volumes with each amplitude played back for ~ 30 s. Manipulations (cutting the wings to mute males, cutting the arista to deafen males) were performed at least 12 h before the start of the experiment under cold anesthesia.

**Sexual satiation assays.** To investigate the effect of sexual satiation, we used a satiation assay (Fig. 5O–Q). For these experiments, we paired a virgin mature male with a satiated male, both of which were 3 to 7 days old. The males were satiated by placing them in the presence of four females for up to 12 h before the experiment. This allowed the males to copulated multiple times before the experiment and induced sexual satiation. To discriminate between the virgin and satiated males, the virgin mature males were marked with a white dot on their thorax using acrylic paint under ice anesthesia 12 h before the experiment. To control for the ice treatment, the males to be satiated were also placed under ice anesthesia before placing them with the females.

## Pose tracking and processing

The position of six body parts (head, neck, thorax, abdomen, tips of the left and right wing) was tracked across frames for each fly using Deepposekit[124]. For most of the analyses, the tracking data was down-sampled from the original frame rate of 100 Hz to 30 Hz. The tracks were transformed into a set of 10 metrics describing the locomotion and relative positioning of the two flies in the chamber. An interaction is defined as when the flies are within a distance of 8 mm and the partner fly is within a field of view of ± 60° of the counter. When both flies are within a field of view of ± 60° of each other (flies facing each other), the fly that was initially the counter remained so. The fly velocities in the x- and y-directions were computed as the rate of change of the fly thorax position along the respective directions. The forward velocities (cFV/pFV; c: counter, p: partner) were computed as the component of the velocity vector in the direction of the fly's body axis (defined by the line connecting head and abdomen). Lateral velocity is the velocity component perpendicular to the body axis. Due to the left-right symmetry, we consider only the magnitude of lateral velocity—the lateral speed (cLS/pLS). The angle of each fly is computed as the angle spanned by the body axis and the image x-axis (positive is counter-clockwise). The rotational velocity of the fly is computed as the rate of change of that angle. Again, due to left-right symmetry, only the magnitude of the rotational velocity, the rotational speed (cRS/pRS), is considered. The distance between two flies (dis) was computed as the Euclidean distance between their thorax positions. The relative angle of one fly with respect to another ($c\theta$ and $p\theta$) is computed as the angle between the body axis of the partner and the line connecting the thoraces of the two flies. The relative orientation ($\phi$) is computed as the difference between the body angles of the two flies. These metrics function as input features to both the HMM-GLM model that predicts the male song patterning as well as the unsupervised social maps.

For detecting unilateral and bilateral wing extensions (Figs. 1C, S3A), the angle of each wing was computed as the angle between the line joining the head and thorax and the line joining the thorax and the corresponding wing tip. A unilateral wing extension occurs when one of the wing angles exceeds a specific threshold. For each trial, this threshold is set at the wing angle that maximizes the number of frames correctly identified as song frames compared to human annotations. A bilateral wing occurs when both the left and right wing angles exceed a threshold of 30°.

**Quantifying counter position during interactions.** We used a polar histogram to quantify the position of the counter around the partner. The distance was measured in terms of the partner fly length. The radial position axis was limited to three fly lengths and binned in 15 equal intervals. Since the side (left or right) on which the counter is placed is not relevant, we used absolute values of relative counter angle $c\theta$ for the angular position. The angular position was thus limited from 0° to 180° and divided into 36 bins of 5° each. $f\theta = 0°$ specified the head and $f\theta = 180°$ specified the tail (abdomen) of the partner. The quadrant spanning $c\theta = 0°$ to $c\theta = 90°$ was considered as the *head* quadrant (the counter was placed near the head of the partner) and the quadrant spanning $c\theta = 90°$ to $c\theta = 180°$ was considered as the *tail* quadrant (the counter was placed near the tail of the partner). To compute the probability of being placed in a quadrant during the interaction, that is P(quadrant|interaction), we ignored the bouts in which the counter was situated in a given quadrant for less than 0.5 s to remove short

transients and possible effects of noise. Similarly, if the time elapsed between the end of a bout and the start of the next bout in the same quadrant was less than 0.5 s, they were considered to be the same bout. After this, the P(quadrant|interaction) was computed as the ratio of frames in which the courter was placed in a given quadrant and the frames in which the courter was interacting with the partner.

**Transitions from tail to head interactions.** We defined a tail interactions as instances when the courter was within 8 mm from the partner fly ($dis < 8$ mm), the partner was within a field of view of 60° ($p\theta < 60°$), and the courter was positioned in the tail quadrant of the partner fly ($90° < c\theta < 180°$). Head interactions were identified when $dis < 8$ mm, $p\theta < 60°$, and the courter was positioned in the head quadrant ($c\theta < 90°$). We identified transitions from tail to head interactions when the courter, after interacting in the tail quadrant for at least two seconds, moved into the head quadrant and remained there for at least two seconds.

We further quantified the probability of song in time windows of different durations leading to transitions to head interactions. For each transition, we considered a window of $L$ seconds before the transition and computed the probability of singing pulse, sine, or either in this window across all such transitions. We compared this with the probabilities of song in the windows that did not lead to a transition. For this, we considered 100 random windows of length $L$ per experiment where the endpoint of the window was at least one minute away from a transition. To ensure that the effect was not linked to the difference in overall interaction between the flies during these windows, we ignored windows in which the interaction ratio was less than 25%.

### Annotation and analysis of the song

Acoustic signals were annotated using Deep Audio Segmenter (DAS,[54]), a deep learning-based tool for annotating multi-channel audio recordings. The acoustic recordings were segmented into four types–pulse song, sine song, agonistic song or wing flicks and noise (silence). To automatically annotate the recordings, the deep learning network was trained on a set of manually annotated recordings. The trained network was used to automatically annotate the remaining recordings which were then manually proofread. When both flies sang simultaneously (during head interactions), they almost always produced a pure sine song. In rare cases, where the song type of flies differed (less than 1% of head interaction time), we attributed the courter's song type as the one he was producing immediately before the onset of mutual singing. Since such mismatches were rare, we do not expect misattribution to significantly impact our results.

We computed signal-level and pattern-level characteristics of the acoustic signals. To characterize pulse song, we extracted individual pulse waveforms from the song recording with a duration of 35 ms centered around each pulse center. The pulse carrier frequency was computed by performing a fast Fourier transform and computing the central frequency in the frequency spectrum with a cutoff at 1000 Hz[41]. The pulse width was calculated based on the signal envelope, which was estimated using the Hilbert transform, and which was convolved with a Gaussian window of width 1.5 ms. The pulse width is then given by the duration the envelope remains above half of its peak amplitude. Pulses with a low signal-to-noise ratio or with multiple peaks were excluded from the analysis of pulse width. The pulse signals whose difference in maximum to mean envelope value is below a threshold of 0.08 was considered noisy and those pulses where the envelope crosses the threshold of 0.5 more than once was considered a multi-peak pulse. The pulse inter-pulse interval (IPI) was computed as the interval between to subsequent pulses. IPIs longer than 100 ms belong to different bouts and were excluded. The carrier frequency of sine song was based on the spectrogram of individual sine waveforms obtained by dividing each sine bout into segments of 256 samples

( ~ 25 ms) and calculating the modal central frequency across segments.

We discriminated the two pulse types $P_{fast}$ and $P_{slow}$ (Fig. S3C, D) using unsupervised clustering of the pulse waveforms. We normalized[41], embedded the pulse waveforms into a UMAP, and clustered them using the H-DBSCAN algorithm[41,57,125]. We considered the two biggest clusters produced by H-DBSCAN, and labeled all pulses within each cluster based on the shape of the waveforms in each of the clusters. The data in the remaining clusters were labeled by training a support vector classifier (SVC[126]) using the labels obtained from the two biggest clusters.

We also computed the amount, the number of onsets, and the duration for pulse trains, sine songs, and all song (combining pulse and sine) in overlapping windows of 1 min (50% overlap). All song bouts were defined as a sequence of pulse trains and sine song that were interleaved by less than 100 ms of silence. The amount was given by the fraction of the window occupied by pulse, sine, or all song. The number of onsets is given by the number of transitions into pulse, sine or any song. The bout duration is computed as the ratio of the amount and the number of onsets in each window. For each quantity, the average over all windows for a given fly was computed.

The bout order is given by one plus the number of song-mode transitions within a song bout. A bout order of 1 corresponds to bouts with only pulse or only sine. A bout with a single transition between pulse and sine has bout order two, and so on.

### Social maps

We constructed social maps by jointly embedding the dynamical tracking features from both interaction partners into a low-dimensional space. For an experiment of duration $T$ and a time history of $K = 15$ samples (0.5 ms) for $M = 10$ features, we created a design matrix of size ($T$, $KxM$) by flattening the time history of all features into a vector. We then fitted a nonlinear dimensionality reduction algorithm, UMAP[57], using 10% of uniformly temporally spaced samples from all experiments of both male-female and male-male interactions to obtain a two-dimensional embedding space. We then embedded the remaining data from all the trials into this low-dimensional space. Thus, for each trial we generate an embedding of size $T \times 2$ where $T$ is the number of samples in a trial. With the data pooled from all the trials, the two-dimensional embedding space functions as a state space that captures the social interactions with similar interactions clustered together and different interactions embedded away from each other. A density function of the samples in this state space characterizes the probability of visiting that region of the state space. To create a continuous state space representation, we perform a kernel density estimation over the samples in the low dimensional representation. Depending on the frequency of certain interactions, this state space consists of regions with higher probability (stereotypic interactions that occur often) and regions with lower probability (for example, rarely occurring interactions or transitions between two stereotypic interactions). Thus a probability-aware segmentation of the interaction state space would generate distinct territories belonging to distinct interaction contexts between the animals. To achieve this, we perform a watershed segmentation over the inverted density estimate of the UMAP-embedded samples[56]. We confirmed by manual inspection that the different watershed segmented territories correspond to stereotypic interaction contexts.

### Maps of behavioral responses to song

To analyze the responses of male and female partners to courtship song, we generated behavioral maps based only on the partners' egocentric tracking features. A frequency spectrogram was computed from each pose coordinate data (normalized using fly length to account for variations in fly size) using a continuous Morelet wavelet

transform which returned the signal power at 25 dyadically spaced frequencies ranging from 1 to 25 Hz. Thus for each fly, we obtain a time series of dimension $T \times 24 \times 25$ where $T$ is the trial duration. This time series is then flattened into a dimension of $T \times 600$ which is then embedded into a two-dimensional space using manifold embedding technique UMAP[57]. As before, we trained the UMAP embedding algorithm using 10% of data (uniformly sampled in time) from all the flies and used the learned UMAP to embed the rest of the data, performed kernel density estimation and partitioned the state-space into stereotypical behaviors using the watershed transform.

## HMM-GLM modeling

We used an HMM-GLM to model the sensorimotor rules that map the feedback cues to the male's choice of song mode (pulse, sine, or silence) at each time point following[13]. An HMM-GLM model is a combination of a hidden Markov model (HMM) and a generalized linear model (GLM). An HMM identifies hidden states from observations (here mode of the male courtship song). In a vanilla HMM, at each time step, the model is in one of the hidden states and has a fixed probability of transitioning to another state or staying in the same state. Similarly, for each state, there is a fixed probability of observing an outcome (a song mode). Although the hidden states identified by the HMM may reveal the internal states of the animal associated with the observed behavior, it does not take into account the effect of external stimuli on the behavior. To account for this, an HMM-GLM associates with each hidden state a GLM that maps the history of sensory cues at each time point to the song mode probability. As surrogate sensory cues, we used the 10 metrics extracted from pose-tracking data. These metrics describe the locomotion and relative positioning of the two flies and thus function as proxies for the feedback cues that a courter male would receive when patterning the song. For each of the 10 metrics, we used a stimulus history of 4 s (120 samples at 30 Hz) as input to the model at each time point. Thus at each time point, the model receives a feature vector of length 1200.

**Hidden Markov model (HMM).** A Markov process or a Markov chain $\mathbf{z}$ is a stochastic process that describes a sequence of states of a system in which the probability of being in a particular state at any given time $t$ depends only on the state occupied in the immediate previous time $t-1$.

In a hidden Markov model (HMM), the stochastic process underlying the sequence of states is not directly observable (hidden), but can only be observed via another stochastic process $\mathbf{y}$ whose outcome depends on the hidden Markov process $\mathbf{z}$. Let $\{z_1, z_2, \ldots z_T\}$ represent a sequence of hidden states of $\mathbf{z}$, and $\{y_1, y_2, \ldots y_T\}$ represent the corresponding observations from $\mathbf{y}$. In our study, the hidden states represent a distinct sensorimotor rule that the male used to pattern his song at each time point, whereas the observation refers to a discrete song mode (pulse, sine or silence) that the male produced at each time point.

An HMM is characterized by the number of discrete hidden states $K$, the number of distinct observations $M$ (here the number of song modes), an initial probability distribution $\boldsymbol{\pi} \in \mathcal{R}^K$ (where $K$ is the number of hidden states), whose each element gives the probability to be in a state at $t = 1$, with $\pi_i = P(z_1 = i)$, a state transition matrix $\boldsymbol{\alpha} \in \mathcal{R}^{K \times K}$ whose elements $\alpha_{i,j} = = P(z_t = j | z_{t-1} = i)$ gives the probability of transitioning from one hidden state $i$ at time $t-1$ to another hidden state $j$ at time $t$, and an observation matrix $\boldsymbol{\eta} \in \mathcal{R}^{M \times K}$ whose each element $\eta_{k,j} = P(y_t = k | z_t = j)$ gives the probability of observing an observation $k$ when the system is in state $j$. Therefore an HMM can be completely specified by its parameters as

$$\Theta = (\pi, \mathbf{A}, \eta) \tag{1}$$

**Generalized linear model (GLM).** To incorporate the effect of external sensory inputs on the male's song patterning behavior, we replaced the

fixed probability observation matrix $\eta \in \mathcal{R}^{M \times K}$ by a generalized linear model (GLM) which maps the history of feedback cues at each time point $\mathbf{s}_{t-N}, \ldots, \mathbf{s}_t$ to a song mode probability $P(y_t = i | \mathbf{s}_{t-N}, \ldots, \mathbf{s}_t)$. Similar to the[13], we used a multinomial GLM which outputs probabilities of three types of song: pulse, sine, and silence at each time point. For simplicity, we considered fast and slow pulses[41] as a single pulse mode. The feedback history $\mathbf{s}_{t-N}, \ldots, \mathbf{s}_t$ is convolved with a set of $D$ basis functions. Each basis function is a vector $\mathbf{b}_j \in \mathcal{R}^N$ of dimension equal to the cue history $N$. We used raised cosine functions[127] that broaden with delay to capture the fact that the effect of cues is often strongest close to the behavioral response (small delays). The basis transformation serves two purposes: (1) It smoothens the feedback cues and thus reduces the effect of noise on the model, and (2) it reduces the dimensionality of model inputs from $10 \times N$ to $10 \times D$ (usually $D \ll N$). Thus the transformed feature vector $\mathbf{x} = \{\mathbf{s}_{t-N}, \ldots, \mathbf{s}_t\} \cdot \{\mathbf{b}_j \in \mathcal{R}^N, j \in \{1, \ldots D\}\}$ has a length of $10 \times D$. This feature vector is z-scored and is augmented with a '1' to incorporate a bias, thus yielding a vector of length $(10 \times D) + 1$ as input to the GLM. The transformed input $\mathbf{x} \in \mathcal{R}^{(10 \times D)+1}$ is passed through linear filters $\mathbf{w}_i \in \mathcal{R}^{(10 \times D)+1}, i \in \{1, 2, 3\}$ corresponding to three song modes, and a softmax function which maps the vector of filtered feedback cues to a normalized probability measure of each song type at each time point $t$.

$$P(\mathbf{y}_t = i | \mathbf{x}_t) = \frac{\exp(\mathbf{w}_i \cdot \mathbf{x}_t)}{\sum_{j=1}^{3} \exp(\mathbf{w}_j \cdot \mathbf{x}_t)} \tag{2}$$

The filters for one song type (here silence) are set to zero such that probabilities of all song types sum to 1. While a previous study that used HMM-GLM[13], used a penalty on the difference of filter coefficients to ensure smoothness, we used basis functions as filters (see also ref. 4) to reduce the dimensionality of the inputs and ensure smoothness. An L2 regularization was applied to the filter coefficients.

**HMM-GLM.** The HMM-GLM we used here associates a GLM parameterized by its filter coefficients $\mathbf{W}_k = \{\mathbf{w}_i \in \mathcal{R}^{(10 \times D)+1}, i \in \{1 \ldots M\}\}$, with each hidden state $k$ of the HMM. Similar to a vanilla HMM, the transition from one hidden state to another follows a fixed probability distribution $\boldsymbol{\alpha} \in \mathcal{R}^{K \times K}$ and an initial state distribution $\boldsymbol{\pi} \in \mathcal{R}^K$. Thus, the parameters of HMM-GLM can be written as

$$\Theta = \{\pi, \alpha, \{\mathbf{W}_k\}_{k=1}^{K}\} \tag{3}$$

The optimal parameters of the model are found by maximizing the log-likelihood of the observed song sequences given the model parameters and the feedback cues.

$$\begin{aligned} \mathcal{L}(\Theta | \mathbf{y}, \mathbf{X}) &\equiv \log \sum_{s \in S_{train}} P(\mathbf{y}_s | \mathbf{X}_s, \Theta) \\ &= \log \sum_{s \in S_{train}} \sum_{\mathbf{z}_s} P(\mathbf{y}_s, \mathbf{z}_s | \mathbf{X}_s, \Theta) \end{aligned} \tag{4}$$

where $S_{\text{train}} \subset S$ represents the sessions (trials) whose data (feedback cues and song) is used for learning the HMM-GLM parameters. The above equation requires summing across all possible state combinations $\mathbf{z}_s$ (paths through the hidden Markov chain) for each trial which is of exponential complexity. The forward-backward algorithm proposed by ref. 128 solves it recursively thus reducing the complexity to linear in $T$. The HMM-GLM was trained using the Expectation-Maximization (EM) algorithm proposed by ref. 129.

**Forward-Backward algorithm.** The forward-backward algorithm was proposed by ref. 128 and provides an efficient way to compute the log-likelihood function in Eq. (4). It involves computing the "forward" and "backward" probabilities. The forward probabilities $a_{i,t}$ give the probability of all observations up to time $t$ and the system is in state $i$ at time

$t$ (trial index $s$ omitted for clarity).

$$a_{i,t} \equiv P(\mathbf{y}_{[1:t]}, z_t = i | \mathbf{X}_{[0:t]}, \Theta) \tag{5}$$

The backward probabilities $b_{i,t}$ give the probability of all future observations up to time $T$, if the state at time $t$ is $i$.

$$b_{i,t} \equiv P(\mathbf{y}_{[t+1:T]} | z_t = i, \mathbf{X}_{[t+1:T]}, \Theta) \tag{6}$$

The forward and backward probabilities can be computed recursively by induction[130], thus reducing the complexity to linear in $T$.

**Expectation-Maximization (EM). E-step:** The E-step involves computing the posterior distribution of the sequence of hidden states given the observations and the model parameters, $P(\mathbf{z}|\mathbf{y}, \mathbf{X}, \Theta)$. From the forward and backward probabilities, we can compute the marginal distribution of the states at every time step and the marginal distribution of the state transitions as

$$\gamma_{i,t} = P(z_t = i | \mathbf{y}, \mathbf{X}, \Theta) = \frac{a_{i,t} b_{i,t}}{\sum_{k=1}^{K} a_{k,T}} \tag{7}$$

and

$$\begin{aligned} \zeta_{i,j,t} &= P(z_t = i, z_t + 1 = j | \mathbf{y}, \mathbf{X}, \Theta) \\ &= \frac{a_{i,t} \alpha_{i,j} b_{j,t+1} P(y_{t+1} | z_{t+1} = k, \mathbf{x}_{t+1}, \mathbf{W}_k)}{\sum_{k=1}^{K} a_{k,T}} \end{aligned} \tag{8}$$

**M-step:** The M-step involves computing the parameters $\Theta$ that maximize the expected log-likelihood given the observations and the posterior state distribution computed in E-step.

$$\begin{aligned} \langle \mathcal{L}(\Theta | \mathbf{Y}) \rangle &= \sum_{i=1}^{K} \gamma_{X_1} \log \pi_i + \sum_{t=1}^{T} \sum_{i=1}^{K} \sum_{j=1}^{K} \zeta_{i,j,t} \log \alpha_{i,j} \\ &+ \sum_{t=1}^{T} \sum_{i=1}^{K} \gamma_{i,t} \log \frac{\exp(\mathbf{W}_{i,y_t} \cdot \mathbf{x}_t)}{\sum_{m=1}^{M} \exp(\mathbf{W}_{i,m} \cdot \mathbf{x}_t)} \end{aligned} \tag{9}$$

The E- and M-steps are iterated until convergence is reached. Since the algorithm is susceptible to getting trapped in local minima giving trivial solutions, we fitted the model multiple times with different initialization of model parameters.

**Chance model.** A chance model gives the probability of observing a song mode $m$, at the current time point $t$, as

$$P(y_t = m | \mathbf{X}_t) = \frac{N_m}{N} \tag{10}$$

where $N_m$ is the number of frames for which song mode $m$ is observed, and $N$ is the total number of frames in the trial. Thus, a chance model always outputs the average probability of observing a song mode and ignores the feedback cues.

**Testing.** Once the HMM-GLM parameters were learned, the model performance was validated by computing the log-likelihood of observations (song sequences) on held-out trials.

$$LL_{\text{heldout}}(\Theta) = \sum_{s \in S \backslash S_{\text{train}}} \log P(\mathbf{y}_s | \mathbf{X}_s, \Theta) = \log \sum_{s \in S \backslash S_{\text{train}}} \sum_{k=1}^{K} a_{s,k,T} \tag{11}$$

The normalized log-likelihood gives the improvement in prediction of song mode compared to a chance model and is obtained as

$$NLL_{\text{heldout}}(\Theta) = \frac{LL_{\text{heldout}}(\Theta) - LL_{\text{heldout}}(0)}{n_{\text{heldout}} \log(2)} \tag{12}$$

where $n_{\text{heldout}}$ is the number of heldout samples and $LL_{\text{heldout}}(0)$ is the log-likelihood obtained from a chance model.

The fitted HMM-GLM can also be used to infer the sequence of hidden states for the complete trial given the observations and feedback cues using the E-step (Eq. (7)) or the Viterbi algorithm[131]. Here, the inferred states refer to the distinct sensorimotor rule that the male used to pattern his song at each time point. Once the hidden state sequence is inferred, it can also be used to predict the probability of each song mode at each time point by applying the filters corresponding to the inferred state to the feedback cues at that time point.

**Sex-agnostic and sex-specific models.** To identify whether the males use the same set of sensorimotor rules to pattern their courtship song towards female and male partners, we fitted three models: (1) a model fitted to female-directed song, $\Theta_f$, (2) a model fitted to male-directed song, $\Theta_m$, and (3) a model fitted to both male- and female-directed song, $\Theta_a$. Further, we validated the performance of sex-specific models on the same sex (held-out trials) and opposite sex and the sex-agnostic models on each sex separately (on held-out trials). For example, the performance of the three models in explaining the female-directed song was given by $LL_f(\Theta_f)$, $LL_f(\Theta_m)$, and $LL_f(\Theta_a)$. If these three likelihoods are similar, a model with sex-specific rules does not explain the observed song patterns better than a model with sex-agnostic rules.

**Feature importance.** To evaluate which feedback cues are relevant for song patterning, we performed a feature importance analysis. This was done by randomly shuffling each of the ten features keeping the other features and song patterning data unchanged and then evaluating the model performance on the shuffled data. Feature importance was computed as the log ratio between the log-likelihood of the model on the shuffled data and the original data. The lower the ratio, the higher the importance of the given feature.

**Effect of counter position and orientation on model predictions.** To evaluate the effect of counter male position and orientation around the partner on HMM-GLM predictions of song mode, we computed the difference in predicted probabilities of pulse and sine song by the model using original and simulated positions and orientations of counter male. For each original sample, we create 180 simulated samples by keeping all the features except either counter relative angle ($c\theta$, for counter position) or partner relative angle ($p\theta$, for counter orientation to partner) and relative orientation ($\phi$) unchanged. We then sweep $c\theta$ (or $p\theta$) from 0° to 180° (1° increments). Correspondingly $\phi$ is set to $180° - c\theta$ (when sweeping $f\theta$) or $180° - p\theta$ (when sweeping $t\theta$). For each simulated sample, the model-predicted probability of pulse and sine song was compared to the predicted probabilities for the original sample. The difference in predicted probabilities is then visualized as a function of the original counter position around the partner.

**UMAP visualization of filtered feedback cues.** We collected the filtered feedback features by applying the learned GLM filters based on the inferred state (rule), $z_t$, and the observation (song mode) $y_t$ to the basis transformed inputs $\mathbf{x}_t$ at each time point $t$. A GLM sums up these filtered feedback cues and passes them through a nonlinearity (logistic function) to output a probability for singing a song mode. Therefore, these filtered features represent a decision space that the model inferred to be relevant for song patterning with boundaries in this space illustrating a change in decision between one song mode versus the other. To visualize such a decision space, we reduced the dimensionality of the filtered feedback cues from $(10 \times D) + 1$ to 2 using UMAP[57]. This analysis showed that similar regions in the decision space map to the same song outputs during both female- and male-directed courtship (Fig. S6G).

**Table 2 | Software and algorithms used**

| Resource | Identifier |
|---|---|
| DeepPoseKit 0.3.6 | https://github.com/jgraving/DeepPoseKit[124] |
| DeepAudioSegmenter (DAS) 0.31 | https://github.com/janclemenslab/das[54] |
| SLEAP 1.3.1 | https://sleap.ai[52] |
| GLM utilities 0.5.2 | https://github.com/janclemenslab/glm_utils |
| scikit-learn 1.0.2 | https://scikit-learn.org[132] |
| HMM-GLM (ssm 0.0.1) | https://github.com/lindermanlab/ssm/ |
| umap-learn 0.5.2 | https://github.com/lmcinnes/umap[57] |
| FFTKDE 1.1 | https://github.com/tommyod/KDEpy |
| scikit-image 0.19 | https://scikit-image.org/[133] |
| python 3.7 | https://python.org |
| Scipy 1.7 | https://scipy.org[134] |
| NumPy 1.21 | https://numpy.org/ |
| pandas 1.3 | https://pandas.pydata.org/ |
| seaborn 0.11 | https://seaborn.pydata.org/[135] |
| matplotlib 3.5 | https://matplotlib.org/[136] |
| statannotations 0.4.4 | https://github.com/trevismd/statannotations[137] |
| pywavelets 1.3 | https://pywavelets.readthedocs.io[138] |
| Affinity Designer 1.10 | https://affinity.serif.com/designer/ |

**Inferring states from optogenetic experiments.** We used the HMM-GLM models fitted to the wild-type song patterning data ($\Theta_a$) to infer the rules used by males during optogenetic experiments.

**Statistical analyses**

All statistical tests were performed using a two-sided Mann–Whitney U (unpaired data) or two-sided Wilcoxon signed-rank (paired) test. A significance level of 0.05 was used.

**Reporting summary**

Further information on research design is available in the Nature Portfolio Reporting Summary linked to this article.

## Data availability

All data supporting the findings of this study are available within the paper, its Supplementary Information or a public repository. Source data are provided with this paper as a Source Data file. Raw experimental data generated in this study have been deposited in the Göttingen Research Online database (https://doi.org/10.25625/PTKYTD). Source data are provided with this paper.

## Code availability

All data analyses were performed using the software listed in Table 2. Code for generating the social maps is deposited at https://github.com/janclemenslab/socialUMAP.

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

## Acknowledgements

We thank Frank Kötting and Stephan Löwe from the ENI workshop for their assistance in designing the behavioral chambers, Kimia Alizadeh for laboratory assistance, Lena Lindner for data acquisition, Gesa Hoffmann, Jan Schöning, Christine Gündner, Rüdiger Ludwig, Matthias Weyl, and Christiane Becker for technical and administrative support. We thank David Anderson, Vivek Jayaraman, André Fiala, Peter Andolfatto, Martin Göpfert, Janelia flylight, and Bloomington Stock Center for providing flies. We thank all members of the Clemens lab, as well as Daniela Vallentin, for feedback on the manuscript. We thank the de Bivort Lab for making their fly clip art publicly available. This work was funded via an Emmy Noether Grant (Project number 329518246) and an ERC Starting Grant (Grant agreement No. 851210) to JC. SM and MN were funded by the IMPRS Neurosciences program of the University of Goettingen.

## Author contributions

Conceptualization—S.R. and J.C. Animals and behavioral experiments—S.R., A.P., S.M., and M.N. Modeling—S.R. Data analysis—S.R., A.P., S.M., and M.N. First draft—S.R. and J.C. Feedback on draft—A.P., S.M., and M.N.

## Funding

## Competing interests

The authors declare no competing interests.
