## [Transparent Peer Review file · Nature Communications]

Sex-specific behavioral feedback modulates sensorimotor processing and drives flexible social behavior

Corresponding Author: Mr Jan Clemens

Version 0:

Reviewer comments:

Reviewer #1

(Remarks to the Author)

Nair and colleagues address the operating principles of sensorimotor transformation that enable flexible behaviors in context-dependent social interactions. Starting with the observation that group-housed *Drosophila* males intensively court other males, the authors compared male courtship behaviors directed at females versus males. Using UMAP to generate social maps, they found differences in interaction dynamics between the two social contexts. In particular, males exhibit more head-to-head interactions in male–male courtship, and these interactions are characterized by closer fly distances and more frequent and louder sine song. Using HMM-GLM (which incorporates behavioral rules/states into the modeling) to predict singing behaviors based on fly movements and distance, the authors found that the model performed similarly well regardless of the social context (male-directed versus female-directed) of the behavioral data on which the model was trained, suggesting that males use the same sensorimotor rules for male- and female-directed singing. The sex-specific song patterns of courting males arise from differences in the feedback provided by male versus female partners: female partners are more idle, leading males to circle and sing from a distance, producing more pulse song; in contrast, male partners often turn to face the courting male, resulting in closer interactions and more sine song. Consistently, neuronal manipulations that force male partners into an idle state or induce female partners to turn are sufficient to reproduce these sex-specific song patterns. The authors further show that social arousal triggered by song perception—likely integrated over time—is a major driver of male-male head interactions. Finally, they investigated the neural basis of song-induced male turning behavior by testing the roles of auditory neurons (vPN1 and pC2l) and social command neurons (P1a and pC1x) using optogenetic activation in male-male interactions. Activation of pC2l or P1a increased head-head interactions, whereas activation of vPN1 or pC1x drove more head-tail interactions. These findings suggest distinct neural pathways that link song perception to male behaviors in different social contexts.

This work is of excellent quality and provides important insights into the behavioral rules and possible neurocomputations underlying context-dependent social interactions. The behavioral analyses and modeling are highly rigorous and informative. The demonstrated role of social feedback in shaping context-dependent social dynamics, while operating under the same behavioral rules, is especially compelling and novel. Although I have a few comments, I have no major concerns and, in principle, I enthusiastically support the publication of this work.

My primary comment regards the interpretation of the optogenetic activation experiments used to probe circuit mechanisms (Fig. 6). While these experiments are informative and the differences in behavioral phenotypes among different circuit elements appear compelling, these results can sometimes be tricky to interpret. For example, the activation phenotypes observed with pC2l and P1a might also reflect a reduced ability to discriminate between the head and tail of the partner male under conditions of heightened arousal. Activating both males may further complicate the interpretation, as males might become locked in a head-head position; therefore, the relative time in head-head interactions may not directly reflect how often the partner males would turn back and drive a head-head interaction. Given that genetic tools for neuronal silencing are available, I wonder if complementary silencing experiments—particularly for pC2l—could further strengthen the conclusions (I understand that P1a inactivation would likely be less informative, as it broadly suppresses courtship).

The authors should make it clear in the main text that the intense male-male courtship was observed in group-housed males. Currently this information is buried in the legend of Fig. 5 and the Methods. Otherwise, a naïve reader may assume that male-male courtship is nearly as common and intense as male-female courtship without considering the raising conditions.

I wonder whether both males may sing simultaneously during male-male interactions. If such events are not uncommon, how are they accounted for when attributing song to the correct male?

Can the authors provide some video clips to illustrate representative behaviors/interactions?

Fig. 6G – make the X-axis consistent with the other panels (left: head-tail; right: head-head).

Reviewer #2

(Remarks to the Author)

The study by Nair et al addresses the role of partner feedback as a sensory cue used by *Drosophila* males during courtship. Social communication requires that senders adjust their behavioral output based on the responses from receivers. However, much of the emphasis in the study of the neural basis of social communication is placed on the signals produced by the sender without considering how behavioral feedback impacts future decisions of the sender. This study elegantly provides detailed behavioral characterization of this process and provides a mechanistic approach to teasing apart the neural underpinnings of this process. It is a very impressive study which will be of broad interest to a broad audience ranging from behaviorists to systems neuroscientists. Its strengths include the impressive technical precision with which the behavioral data has been collected, the robust quantitative analysis, use of modeling to identify key features of social interactions and use of identified neurons to determine the cellular underpinnings of key behavioral choices. The methodology is sound and sufficiently detailed in its description to be replicated by others. My comments are only suggestions to further improve ease of interpretation for the reader and I do not see the need for additional experimentation which I fear would overwhelm an already data-rich manuscript. I commend the authors for a truly impressive study.

Comments:

-It would be helpful to include 1-2 sentences when first introducing the social maps so that the reader better understand the nature of this form of representation. What is the significance of the shape of each territory, the amount of boundary shared between territories, etc.

-The authors found that for male-male head-to-head interactions there is an increase in the amount of song produced (specifically sine song). To my understanding, they also found that by several metrics (Figure 2B) that the nature (variability, carrier frequency) of the song doesn't seem to differ, only duration. Personally, I think that this is an interesting observation because it is supportive of the idea that a limited set of behaviors can be used in different combinations, rather than the need for a sex specific song. However, I defer to the authors as to whether this is worth further discussion.

-For Figure 4G, it could help with interpretation to indicate that these are immobilized flies, as the authors indicate that pC1d is activated in figure 4K.

-For the text in the results describing figure 5, a bit more description would make it easier for the reader to follow the manipulations. As it stands it is a bit confusing as to which fly is deafened or muted in which combination and when song is being played for the flies as opposed to produced by the flies in the arena. Sentences along the lines of "to determine XYZ, we exposed muted or deafened flies to recordings of male song played into the arena". This is particularly true for the sexual satiation experiments which are barely mentioned in the results section, without a description of the findings. Overall, this needs to be fleshed out a bit more as it is too dense to give the experiments their proper credit.

Version 1:

Reviewer comments:

Reviewer #1

(Remarks to the Author)

The authors have rigorously addressed all of my comments, including the addition of neuronal inactivation experiment to strength the conclusion. This work compellingly demonstrates how social feedback alone can shape context-dependent social dynamics using the same behavioral rules. It is of broad significance, and I enthusiastically support its publication.

(Remarks on code availability)

We thank the Reviewers for their positive and insightful feedback. We are thankful that the Reviewers found our work of excellent quality and rigour and of broad interest to the community. Here we provide point-by-point responses to the Reviewers' comments along with the corresponding changes in the manuscript.

Reviewer comments are formatted in blue font, our replies in black font, citations from the paper text in italics.

REVIEWER COMMENTS

Reviewer #1 (Remarks to the Author):

Nair and colleagues address the operating principles of sensorimotor transformation that enable flexible behaviors in context-dependent social interactions. Starting with the observation that group-housed *Drosophila* males intensively court other males, the authors compared male courtship behaviors directed at females versus males. Using UMAP to generate social maps, they found differences in interaction dynamics between the two social contexts. In particular, males exhibit more head-to-head interactions in male–male courtship, and these interactions are characterized by closer fly distances and more frequent and louder sine song. Using HMM-GLM (which incorporates behavioral rules/states into the modeling) to predict singing behaviors based on fly movements and distance, the authors found that the model performed similarly well regardless of the social context (male-directed versus female-directed) of the behavioral data on which the model was trained, suggesting that males use the same sensorimotor rules for male- and female-directed singing. The sex-specific song patterns of courting males arise from differences in the feedback provided by male versus female partners: female partners are more idle, leading males to circle and sing from a distance, producing more pulse song; in contrast, male partners often turn to face the courting male, resulting in closer interactions and more sine song. Consistently, neuronal manipulations that force male partners into an idle state or induce female partners to turn are sufficient to reproduce these sex-specific song patterns. The authors further show that social arousal triggered by song perception—likely integrated over time—is a major driver of male-male head interactions. Finally, they investigated the neural basis of song-induced male turning behavior by testing the roles of auditory neurons (vPN1 and pC2I) and social command neurons (P1a and pC1x) using optogenetic activation in male-male interactions. Activation of pC2I or P1a increased head-head interactions, whereas activation of vPN1 or pC1x drove more head-tail interactions. These findings suggest distinct neural pathways that link song perception to male behaviors in different social contexts.

This work is of excellent quality and provides important insights into the behavioral rules and possible neurocomputations underlying context-dependent social interactions. The behavioral analyses and modeling are highly rigorous and informative. The demonstrated role of social feedback in shaping context-dependent social dynamics, while operating under the same behavioral rules, is especially compelling and novel. Although I have a few comments, I have no major concerns and, in principle, I enthusiastically support the publication of this work.

My primary comment regards the interpretation of the optogenetic activation experiments used to probe circuit mechanisms (Fig. 6). While these experiments are informative and the differences in behavioral phenotypes among different circuit elements appear compelling, these results can sometimes be tricky to interpret. For example, the activation phenotypes observed with pC2I and P1a might also reflect a reduced ability to discriminate between the head and tail of the partner male under conditions of heightened arousal. Activating both males may further complicate the interpretation, as males might become locked in a head-head position; therefore, the relative time in head-head interactions may not directly reflect how often the partner males would turn back and drive a head-head interaction. Given that genetic tools for neuronal silencing are available, I wonder if complementary silencing experiments—particularly for pC2I—could further strengthen the conclusions (I understand that P1a inactivation would likely be less informative, as it broadly suppresses courtship).

Thank you for this suggestion. We now show that neuronal silencing of pC2 neurons reduces turning into head interactions, strengthening our argument regarding the neuronal underpinnings of this behavior.

To address this issue, we silenced pC2 neurons (R41B02 Gal4) using tetanus toxin (TNT) in males. Pairs of pC2-TNT males as well as the associated genetic controls (pC2-TNTinactive) had very low baseline interaction rates (probably because of a genetic background effect). To increase baseline interaction rates, we paired pC2-TNT males with P1a-activated males (new Fig. 6 panels K–O, reproduced below). We find that pC2 inactivation reduces, but does not abolish, the turning into head interactions, demonstrating that pC2 neurons contribute to inducing the head-to-head context, but are not necessary for it.

We have included these results in the revised manuscript in Fig. 6 panels K–O. Since Fig. 6 was already full page, we've split Fig. 6 (old Fig. 6A–J, along with new panels K–O is now Fig. 6, old Fig. 6K–Y is now Fig. 7). We describe the new experiments on p. 10, lines 262–267: *To show that pC2 neurons contribute to head interactions driven by turning responses, we silenced pC2 neurons using tetanus toxin (TNT) (Fig. 6K). A pC2-silenced or control (TNT-inactive) male was paired with a P1a-activated male to induce high levels of male-male interactions. The pC2-silenced males initiated fewer head interactions and turned less towards the P1a-activated courter compared with control males (Fig. 6L–N). In addition, head interactions were shorter in pC2-silenced males (Fig. 6O).*

As pC2 silencing does not completely abolish the partner turning responses in males, we now discuss the contribution of other song detection neurons in the Discussion, p. 14, lines 349–351:

Silencing of pC2 neurons did not completely abolish the head interaction between males (Fig. 6L–N), suggesting that additional song detection neurons such as pC1 (Zhou et. al. 2014, Zhou et. al. 2015) can induce head interactions.

The details of the pC2 silencing experiments and analysis are provided in the “Methods” section, lines 459–466.

New panels K–O in Fig. 6:

K Experimental procedure for pC2 silencing. pC2-silenced males are paired with p1a-activated males (males expressing csChrimson in p1a neurons were exposed to red light for the whole duration of the trial to increase their intensity of interaction towards a male partner).

L Fraction of trial time spent by p1a-activated male near the head and tail of pC2-silenced and control males.

M Position of p1a-activated male around pC2-silenced and control males.

N Fraction of interaction time spent by p1a-activated male in the head and tail quadrants of pC2-silenced and control males.

O Dwell times of p1a-activated male in the head and tail quadrants of pC2-silenced and control males. Dots in L, N and O correspond to averages for each fly pair (control: N=21 pairs, silencing: N=18 pairs).

For panels L, N, and O, p-values are computed using two-sided Mann-Whitney-U tests. Dots correspond to fly pairs.

The authors should make it clear in the main text that the intense male-male courtship was observed in group-housed males. Currently this information is buried in the legend of Fig. 5 and the Methods. Otherwise, a naïve reader may assume that male-male courtship is nearly as common and intense as male-female courtship without considering the raising conditions.

Thank you for this suggestion. We have added the following line in the text describing the experiments in the first section of results (p. 3, lines 88–90):

To promote male-male interactions, we controlled the males' prior social experience by raising them in groups and ... In our assay, group-housed males interacted intensely with both males and females ...

I wonder whether both males may sing simultaneously during male-male interactions. If such events are not uncommon, how are they accounted for when attributing song to the correct male?

Yes, both males occasionally sing simultaneously during head-to-head interactions. However, simultaneous singing was rare and therefore did not impact our conclusions. We have updated the Methods subsection “Annotation and analysis of the song” to clarify our attribution procedure (Lines 558–562):

When both flies sang simultaneously (during head interactions), they almost always produced a pure sine song. In rare cases, where the song type of flies differed (less than 1% of head interaction time), we attributed the courter's song type as the one he was producing immediately before the onset of mutual singing. Since such mismatches were rare, we do not expect misattribution to significantly impact our results.

Can the authors provide some video clips to illustrate representative behaviors/interactions?

We have added representative videos of the social modes as Supplementary material and refer to them in the text describing the social maps in Fig. 1. Additionally, we also provide video snippets of transitions from tail to head interactions towards both females and males to supplement the results shown in Fig. 4C.

Fig. 6G – make the X-axis consistent with the other panels (left: head-tail; right: head-head).

Thank you for pointing this out. We have corrected this.

Reviewer #2 (Remarks to the Author):

The study by Nair et al addresses the role of partner feedback as a sensory cue used by *Drosophila* males during courtship. Social communication requires that senders to adjust their behavioral output based on the responses from receivers. However, much of the emphasis in the study of the neural basis of social communication is placed on the signals produced by the sender without considering how behavioral feedback impacts future decisions of the sender. This study elegantly provides detailed behavioral characterization of this process and provides a mechanistic approach to teasing apart the neural underpinnings of this process. It is a very impressive study which will be of broad interest to a broad audience ranging from behaviorists to systems neuroscientists. Its strengths include the impressive technical precision with which the behavioral data has been collected, the robust quantitative analysis, use of modeling to identify key features of social interactions and use of identified neurons to determine the cellular underpinnings of key behavioral choices. The methodology is sound and sufficiently detailed in its description to be replicated by others. My comments are only suggestions to further improve ease of interpretation for the reader and I do not see the need for additional experimentation which I fear would overwhelm an already data-rich manuscript. I commend the authors for a truly impressive study.

Comments:

-It would be helpful to include 1-2 sentences when first introducing the social maps so that the reader better understand the nature of this form of representation. What is the significance of the shape of each territory, the amount of boundary shared between territories, etc.

We thank the Reviewer for this suggestion. We have updated the results and methods section to briefly describe the nature of the social map representations as per your suggestion.

In the results section, we now state in p. 3, lines 109–116:

“... we generated social maps using UMAP, which embeds the egocentric and relational kinematics of both flies into two dimensions (Fig. 1E–F, S2) [58–61]. In these maps, similar interaction patterns are positioned close together, whereas dissimilar patterns are separated. High-density regions correspond to stereotypical interaction patterns (social modes), while

low-density regions indicate transitions between modes. Because UMAP is a nonlinear embedding method, distances and the shapes of high-density regions are not directly interpretable. The embedding space was therefore smoothed using kernel density estimation and segmented into social modes using watershed-based spatial segmentation.

-The authors found that for male-male head-to-head interactions there is an increase in the amount of song produced (specifically sine song). To my understanding, they also found that by several metrics (Figure 2B) that the nature (variability, carrier frequency) of the song doesn't seem to differ, only duration. Personally, I think that this is an interesting observation because it is supportive of the idea that a limited set of behaviors can be used in different combinations, rather than the need for a sex specific song. However, I defer to the authors as to whether this is worth further discussion.

This is an excellent suggestion, and as you rightly pointed out, it fits in nicely with the main message of our paper: flexible and complex behavior can arise from a limited set of behavioral primitives by using them in different combinations depending on the context (compositionality). We now include a brief discussion of these results in the section in Results, lines 143–147, as follows:

Together, these results show that while males use the same song signals when courting both female and male targets, they combine these signals differently depending on the social context, leading to sex-specific song patterning. Thus, song patterning in flies exhibits a compositional structure, whereby complex behaviors arise from a limited set of behavioral primitives that are recombined according to context [64].

Furthermore, we edit the discussion on compositionality, p. 14, lines 378–385:

Our results further show that in flies, compositionality exists not only at the level of rules (input-output mapping), but also in the sequencing of motor outputs [64, 115, 116]. The sex-specific song sequences towards females and males do not arise from differences in the characteristics of the song signal itself (carrier frequency, width, duration, IPI), but from males patterning the same signals in different combinations towards the two sexes, depending on the interaction context (Fig. 2). More broadly, our findings highlight how flexible, context-dependent behaviors can emerge from a static rule set and limited behavioral repertoire via feedback-driven modulation, proposing a mechanism for flexibility of innate behaviors.

-For Figure 4G, it could help with interpretation to indicate that these are immobilized flies, as the authors indicate that pC1d is activated in figure 4K.

We agree with your suggestion. We have now updated Figure 4G to indicate that the targets are immobilized.

-For the text in the results describing figure 5, a bit more description would make it easier for the reader to follow the manipulations. As it stands it is a bit confusing as to which fly is deafened or muted in which combination and when song is being played for the flies as opposed to produced by the flies in the arena. Sentences along the lines of “to determine XYZ, we exposed muted or deafened flies to recordings of male song played into the arena”. This is particularly true for the sexual satiation experiments which are barely mentioned in the results section, without a description of the findings. Overall, this needs to be fleshed out a bit more as it is too dense to give the experiments their proper credit.

We thank the Reviewer for pointing this out. We have revised the Results section describing Fig. 5 to provide a clearer context for each manipulation. We now explicitly state the following in the manuscript on p. 7, p. 10, lines 215–232:

To test whether courtship song drives the turning of the male partner, we selectively disrupted song production or perception in male pairs. We paired either a muted (wingcut, Fig. 5A) or a deaf (aristacut, Fig. 5E) with an intact partner capable of both singing and hearing. Disrupting acoustic communication strongly reduced head interactions (Fig. 5A, E right) and the few remaining head interactions were consistently initiated by the male capable of perceiving song.

In pairs containing a muted male, courtship by the muted male did not expose the intact male to song and accordingly courtship from the mute male predominantly led to tail interactions. In contrast, courtship from the intact male exposed the muted male to song and led to head interactions, as the muted male turned toward the intact male in response to song (Fig. 5B–D). Similarly, in pairs containing a deaf male, courtship from the intact male toward the deaf male resulted mainly in tail interactions, as the deaf male was not able to perceive the intact male's song. Courtship from the deaf male exposed the intact male to song and led to head interactions induced by the intact male turning back in response to the deaf male's song (Fig. 5F–H).

Thus, perception of song is necessary for turning responses that generate head interactions. To test sufficiency, we exposed a pair of muted (wingcut) males to playback of recorded pulse song (Fig. 5I, left). Song playback elicited turning responses in the male being courted and increased head interactions (Fig. 5I–L). Together, these experiments demonstrate that perceiving song is both necessary and sufficient for head interactions.”

Concerning the sexual satiation experiments, we now state the following in p. 10, lines 242–246:

To further investigate the role of the partner's sexual arousal in inducing head interactions, we paired sexually satiated males with virgin mature males. The head interactions were rare in sexually satiated males with low sexual arousal compared to virgin males (Fig. 5P--Q). These results illustrate that the perception of courtship song combines with a heightened internal arousal state to drive head interactions in males.